# Activity Recognition for Ambient Assisted Living with Videos, Inertial Units and Ambient Sensors

**DOI:** 10.3390/s21030768

**Published:** 2021-01-24

**Authors:** Caetano Mazzoni Ranieri, Scott MacLeod, Mauro Dragone, Patricia Amancio Vargas, Roseli Aparecida Francelin Romero 

**Affiliations:** 1Institute of Mathematical and Computer Sciences, University of Sao Paulo, Sao Carlos, SP 13566-590, Brazil; cmranieri@usp.br; 2Edinburgh Centre for Robotics, Heriot-Watt University, Edinburgh, EH14 4AS, UK; sam19@hw.ac.uk (S.M.); M.Dragone@hw.ac.uk (M.D.); p.a.vargas@hw.ac.uk (P.A.V.)

**Keywords:** human activity recognition, multimodal datasets, deep learning, video classification, inertial sensors, human–robot interaction

## Abstract

Worldwide demographic projections point to a progressively older population. This fact has fostered research on Ambient Assisted Living, which includes developments on smart homes and social robots. To endow such environments with truly autonomous behaviours, algorithms must extract semantically meaningful information from whichever sensor data is available. Human activity recognition is one of the most active fields of research within this context. Proposed approaches vary according to the input modality and the environments considered. Different from others, this paper addresses the problem of recognising heterogeneous activities of daily living centred in home environments considering simultaneously data from videos, wearable IMUs and ambient sensors. For this, two contributions are presented. The first is the creation of the Heriot-Watt University/University of Sao Paulo (HWU-USP) activities dataset, which was recorded at the Robotic Assisted Living Testbed at Heriot-Watt University. This dataset differs from other multimodal datasets due to the fact that it consists of daily living activities with either periodical patterns or long-term dependencies, which are captured in a very rich and heterogeneous sensing environment. In particular, this dataset combines data from a humanoid robot’s RGBD (RGB + depth) camera, with inertial sensors from wearable devices, and ambient sensors from a smart home. The second contribution is the proposal of a Deep Learning (DL) framework, which provides multimodal activity recognition based on videos, inertial sensors and ambient sensors from the smart home, on their own or fused to each other. The classification DL framework has also validated on our dataset and on the University of Texas at Dallas Multimodal Human Activities Dataset (UTD-MHAD), a widely used benchmark for activity recognition based on videos and inertial sensors, providing a comparative analysis between the results on the two datasets considered. Results demonstrate that the introduction of data from ambient sensors expressively improved the accuracy results.

## 1. Introduction

According to projections by the Department of Economic and Social Affairs of the United Nations, the worldwide proportion of citizens aged between 15 and 64, with respect to those aged over 65 years old, is expected to drop from about 7:1 in 2020 to approximately 4:1 in 2050 [1]. This may lead to a deficit in workforce numbers in the elderly care sector, which has motivated the research on Ambient Assisted Living (AAL) [2]. The idea is to support human carers, with the introduction of assistive technologies. These solutions may help to address issues such as improving limitations of movements, monitoring chronic diseases, minimising social isolation or controlling medicine administration by providing integrated services that may be connected to the Internet of Things (IoT) [3].

Technologies for AAL may be provided in the form of smart homes [4], equipped with sensors, for monitoring different conditions of the environment and its inhabitants [5], and actuators, to effectively help them in their daily activities [6]. To enhance those environments and improve their acceptance towards the end users, the design can include service or social robots [7], which may either introduce additional functionalities and monitoring tools, or provide more natural human–robot interaction. One advantage of introducing such robots to an AAL environment is the possibility to collect visual information with less privacy concerns than those related to fixed cameras [8]. Besides, robots may be endowed with behaviours to manage privacy-sensitive situations [9].

Human activity recognition, which consists of classifying human-centred data from different sensors [10], is a key requirement for AAL applications, as it is essential for allowing proactive behaviours or even basic cooperation between human and the environment. The review provided in Chaaraoui et al. [11] presented a discussion on taxonomies for Human Behaviour Analysis (HBA). According to the authors, an *activity* is a sequence of semantically meaningful actions involving interactions between humans and their environment. The most widely adopted approach to HBA involves the classification of the activities from sensor data capturing sequences of basic human motions, i.e., *action primitives*.

To date, most research on this field has focused on single modality approaches, which may consist of either RGB [12] or RGB-D videos [13], wearables such as inertial sensors (Inertial Measurement Units—IMUs) [14], or ambient sensors [15]. The scenarios in which each of these modalities have been employed for activity recognition vary according to the availability of data, which may be constrained by technical or ethical limitations. RGB videos can be found on different online sources, which allows the gathering of different large-scale, very heterogeneous datasets [16]. Depth videos and IMU data are usually collected in more controlled environments, such as AAL research laboratories [17]. For all of those modalities, deep learning (DL) approaches have shown to provide state-of-the-art classification results [18,19,20]. In the case of ambient sensors, most datasets provides long-term records of binary data, and the associated research effort usually focus on segmenting and classifying human activities [21].

The availability of data from multimodal sources within a smart robotic environment [22] may help designing more robust methods for activity recognition. For instance, although recent advances on DL approaches have made video-based activity recognition a very powerful approach [23], this modality of data may be unavailable due to privacy restrictions, or it may be compromised by technical issues such as occlusions. Besides, one modality of data can perform better than another in certain conditions. Ambient sensors may be quite informative on some well-defined scenarios in a smart home [24], while wearable sensors can be more suitable for actions that rely on limb motions [25]. Therefore, most recently, multimodal approaches for activity recognition have been investigated [20] as more robust alternatives when compared to single-modality approaches.

To the best of our knowledge, there is no work in the literature that addressed the problem of recognising heterogeneous activities of daily living centred in home environments by building modules that consider, simultaneously, data from videos, wearable IMUs, and ambient sensors. One of the reasons is the lack of a representative dataset suitable for this task, which would be a prerequisite to train and test any data-driven model. Nonetheless, this configuration can be expected in smart AAL environments combining smart home and robotic technology.

Driven by this motivation, our first contribution in this work is the design, collection and curation of the Heriot-Watt University/University of Sao Paulo (HWU-USP) activities dataset, which will be made public. This database was built based on an international collaboration between researchers from the Heriot-Watt University (HWU) in the UK, and the University of Sao Paulo (USP) in Brazil (the dataset is available at https://drive.google.com/drive/folders/1Aq1kOcAxLhZl84R9qAdW_o0uL8s5b30E?usp=sharing). The dataset was designed to capture a set of activities of daily living that took place in the Robotic Assisted Living Testbed (RALT) at the Heriot-Watt University, in Edinburgh, Scotland. It includes not only activities that involve long-term dependencies, such as preparing a sandwich, but also static activities, such as reading a newspaper. Videos were recorded from the RGBD camera from a robot, positioned at a fixed location in the test kitchen. Two wearable IMUs were placed at the dominant arm and at the waist of each participant, exemplifying the inertial sensors usually found in smartwatches and a smartphones. The ambient sensors from the smart home also have been integrated in the environment.

Besides presenting the dataset in detail, our second contribution is the development of a framework based on Deep Learning (DL) networks for classifying multimodal data not only from videos and inertial units, as performed on related work, but also on ambient sensors. To the best of our knowledge, this is the first approach to consider those three modalities altogether, which could not be done with the other datasets present on the literature. The DL models for the different modalities were trained and evaluated with the HWU-USP dataset. Our investigation included approaches for sensor fusion, a non-trivial problem which drives research in different contexts [26], and has been explored in the field of activity recognition [27]. On our case, fusion was performed mostly at decision-level, though one feature-level approach was proposed for the inertial and ambient sensors. A comparative analysis of the results, quantifying the improvements achieved by each approach, was performed.

The classification framework was based on existing literature for each modality. Regarding the video modality, we have considered the two streams proposed by Simonyan and Zisserman [28]: the spatial and temporal streams. As expected, due to the motion-driven aspect of the datasets analysed, with few background information or objects that could be discriminative regarding to the activity being performed, the appearance-based approaches (i.e., the spatial stream) led to poor results, and hence were not considered on the multimodal scenarios. Instead, our architecture focused on motion-based approaches (i.e., the temporal stream), which led to the best accuracies observed for the single-modality approaches. This consisted of combining CNN modules for feature extraction on dense optical flow maps [29,30,31], previously computed on the video frames, and a LSTM layer for temporal modelling [32].

With respect to the IMU, we introduced the raw, time-domain data to a DL architecture, another common practice in related work [33,34]. The fusion between IMU data and ambient sensors was performed internally as part of one of the DL architectures presented, after both modalities were temporally aligned in a preprocessing stage, an approach that we are proposing as part of this work. To perform fusion between the video-based models and the models that processed IMU and ambient sensors’ data, the output vectors were combined with the outputs of the other modalities, also an approach commonly adopted in related research [35,36].

All predictions are performed on two-seconds-long segments. Following a widely adopted approach in the literature in video-based activity recognition [18,28,35], we have evaluated our models on 25 segments equally spaced between them. We did the same for the other modalities as well, since this approach allows the classifiers to consider partial observations of the activities, as expected for real-world scenarios. Results are presented in terms of the accuracy obtained in each of the conditions analysed, corresponding to different input modalities or fused models. The introduction of ambient sensors has shown to provide significant improvements to the overall accuracy. The results presented here provide a baseline for future work in human activity recognition using multi-modal sensor data in smart robotic environments.

Besides the new HWU-USP dataset, we have also experimented our video and IMU models with another popular public available dataset, the UTD-MHAD [37], providing comparisons with the HWU-USP dataset regarding to the behaviour of the classifiers. Moreover, the classification methods achieved competitive results for the UTD-MHAD. The confidence in predicting the correct label on each segment was also analysed. As was expected, this was quite different when comparing the HWU-USP dataset, consisting of both complex and simple activities, to a more homogeneous dataset, such as the UTD-MHAD.

The remainder of this article is organised as it follows. Section 2 illustrates and compares the most relevant datasets from the literature, and highlights their key differences from the one presented in this paper. Section 3 provides an overview of sensor-based human activity recognition, focused on techniques able to exploit different input modalities. Section 4 presents a detailed description of the data in the proposed dataset and the protocol used for its collection. Furthermore, it describes the DL methods considered for the classification of data and also the protocols used for their training and evaluation. The results are then presented and analysed in Section 5, and a discussion is presented in Section 6. Finally, in Section 7, conclusions and possible directions for further research are outlined.

## 2. Datasets of Human Activities

The HWU-USP dataset was built to provide a benchmark for studies on activity recognition in indoor environments. For this reason, combinations of different modalities, namely videos, wearable IMUs, and environmental sensors were considered. In this section, previously developed datasets that includes sensor data from these modalities, regardless of the context, will be presented, in order to contextualise the construction of the HWU-USP dataset. The nature of available datasets and associated approaches for data collection vary greatly for different sensor modalities considered in human activity recognition research. For example, for RGB video datasets, there is a vast availability of data on the Internet, from movies or other non-dedicated sources, which can be labelled and made available, resulting in fairly large datasets. This is more difficult for depth videos, IMUs or environmental sensors, hence this type of datasets are more often collected in controlled settings, usually in research laboratories simulating domestic environments. In the next subsections, datasets for each modality or set of modalities will be presented separately.

### 2.1. RGB Videos

As already mentioned, most commonly used benchmarks of regular RGB videos can avail of amateur videos, movies or sports broadcasts. Most of these datasets are pre-segmented, which means that each video is entirely associated to one category (e.g., “biking” or “playing piano”), with a few exceptions. The categories in which the activities of these datasets are usually labelled are generally at a comparatively high level of abstraction and granularity, including activities such as *playing basketball*, instead of low-level, primitive activities such as *walking* or *running*. A summary of representative RGB video datasets is provided in Table 1.

The two most relevant benchmarks, on which the most renowned video-based HAR techniques have been evaluated, are the UCF101 [16], from the *University of Central Florida*, and the *Human Motion Database* (HMDB51) [38]. The *Columbia Consumer Video Database* (CCV) [39] is also commonly referenced, as it presents similar properties, but longer videos. The Hollywood2 [40] and the Sports-1M [41] datasets present an additional challenge, as the videos contain editions and camera transitions. Although, as seen in Table 1, the Sports-1M dataset is quite large, a newer dataset—the Kinetics dataset [42]—has been preferred for testing DL architectures, requiring a large amount of data. Regarding datasets that were not pre-segmented, some of the most relevant ones are the THUMOS [43] dataset, provided with the same set of categories as the UCF101, and the ActivityNet [44], annotated according to a semantic hierarchy of activities designed by the U.S. Department of Labour to perform the American Time Use Survey (ATUS).

All of the above-mentioned datasets consist of heterogeneous and realistic sets of videos, usually thanks to user-created content. This variety of data is not possible, at least at present, for data from other modalities, such as RGB and depth videos, wearable and environmental sensors. Consequently, multimodal datasets are usually collected in controlled environments, mostly with static backgrounds, few variations in camera angles and artefacts shared among the data samples. These limitations are inherent to any dataset consisting of modalities that does not count on large amounts of user-created content, which is the case for almost all multimodal datasets, including ours.

### 2.2. Depth Videos

With the popularisation of RGBD (RGB + depth) cameras, such as the Microsoft Kinect [45], it became possible to provide not only RGB and depth videos, but also previously extracted skeleton joints from humans being observed. The categories within these datasets are usually from levels of abstraction compatible with those that could be acquired by RGBD devices, although less diverse, with several activities sharing the same background, objects for manipulation and light conditions. In Table 2, a collection based on the datasets adopted by Amir Shahroudy et al. [46] is shown. These datasets presented were collected using a Microsoft Kinect device, except for the NTU RGB+D, which was collected using a Microsoft Kinect v2. Both devices may collect data on either 15 Hz or 30 Hz.

The datasets listed at Table 2 share a lot of common points. The Online RGBD Action dataset (ORGBD) [47] contains videos from different environments, allowing *cross-environment* evaluation of HAR techniques. The MSR-DailyActivity3D [48] is characterised by a higher intra-class variation. The 3D Action Pairs [49] was designed to include pairs of opposite activities, such as *pull a chair* and *push a chair*. An initiative for providing a larger dataset resulted on the RGBD HuDaAct [50]. Finally, the NTU RGB+D was extended and formed the NTU RGB+D 120 dataset [51], with more than 100K videos distributed on 120 categories.

### 2.3. Wearable and Ambient Sensors

In this subsection, we are addressing sensors that may be worn by the subjects (i.e., wearable sensors) or placed at predefined locations of the environment (i.e., ambient sensors). We focused our review in inertial measurement units (IMU), since most multimodal datasets address this modality. However, we also referenced setups including sensors embedded in the environment, usually at fixed locations, because these can help to get very discriminative information. This is the case of our own dataset, which includes data from different sensors from a smart home, as discussed in Section 4.1. The data provided by these devices usually consist of measurements from accelerometers, gyroscopes, and, sometimes, magnetometers, all of them three-dimensional. All datasets examined in Table 3 were collected under controlled conditions, with the sensors placed on the surfaces of objects or, most commonly, as wearable devices.

The OPPORTUNITY [52] dataset has been widely used as benchmark in the literature for activity recognition tasks involving wearable or environmental sensors, as it consists not only of several inertial sensors placed in objects of daily living and worn by the subjects, but also tags and switches positioned in different parts of the environment. Another widely adopted dataset is the *Physical Activity Monitoring for Aging People* (PAMAP) and its extension, the PAMAP2 [53], designed for identifying patterns in subjects performing physical exercises. The *Realistic Sensor Displacement Benchmark Dataset* (REALDISP) [54] also addresses physical activities. The positioning and availability of sensors are not usually practical and intended for large-scale adoption, except when dealing with standardised conditions, such as smartphones, as addressed on the *Smartphone-Based Human Activity Recognition dataset* (SBHAR) [17]. Datasets for other scenarios have also been developed, such the *Skoda Mini Checkpoint dataset* [55], composed of work activities in a car factory, and the *Daphnet Gait* (DG) [56], composed of motion patterns of patients affected by Parkinson’s Disease.

Bakar et al. [57] presented an extensive survey on sensing approaches for activity recognition in smart homes. Besides cameras, microphones and wearables, these environments allow the introduction on fixed sensors such as temperature, pressure or motion sensors. Binary sensors, such as switches at doors and wardrobes, are also usual, and these categories were also included on our approach. The CASAS project [58] proposed different testbeds that could be used for data collection and experiments in smart homes, based mostly on environmental sensors. Differently from the datasets mentioned in Table 3, these datasets usually result from long-term data collections. As detailed by Lesani et al. [59], the Twor2009, Tulum2009 and Tulum2010 datasets, from the CASAS project, were collected in periods ranging from 3 to 6 months, in which information from motion, binary, door and item sensors were recorded.

An intrinsic advantage of the above-mentioned modalities is that they can provide additional data that are invariant to the positioning of externally placed observing devices, contrary to when cameras or robots are used. Thus, they may provide valuable information for an activity recognition framework. Besides, inertial and ambient sensors also have the advantage of being less intrusive than video cameras. For this reason, multimodal datasets, including video, IMUs and other modalities, have been proposed.

### 2.4. Multimodal: Video and IMU

Multimodal datasets with videos and other sensors, especially IMUs, have been proposed in different contexts. Most of these datasets report data from combinations of different sensors and depth videos, which may be accompanied by the RGB videos. A survey on the subject was provided by Chen et al. [27], considering only datasets that provided depth videos and IMU data. In Table 4, we present a collection of the most relevant datasets for any kind of video collected along with data from other sensors.

The *Carnegie Mellon University Multimodal Activity* (CMU-MMAC) Database [60] records data from RGB cameras, microphones and wearable sensors worn by a set of subjects performing food in a kitchen environment. The *Berkeley Multimodal Human Action Database* (Berkeley-MHAD) [61] and the *University of Texas at Dallas Multimodal Human Action Database* (UTD-MHAD) [37] have a similar structure based on short actions recorded with RGBD cameras, IMUs and a variety of other sensors. The recently deployed Continuous Multimodal Human Action Dataset [62] was collected in an environment similar to that of the UTD-MHAD, although without the RGBD camera, containing activities related to smart TV gestures (5 activities) and transitions (7 activities) in continuous, non-segmented recording sessions. The *50 Salads* dataset [63] captures people preparing several salad recipes being recorded by RGBD cameras, and IMUs placed in the utensils used for the food preparation. The ChAirGest [65] dataset focuses on gesture recognition with the aim to be applied in human–computer interfaces. The *University of Rzeszow Fall Detection Dataset* [66] and the *Telecommunications System Team* (TST) *Fall Detection Dataset* [67] were built with data on regular daily activities and falls, which can be used to train models for fall detection, an important field of research with applications as part of AAL solutions for the elderly.

Although the above-mentioned datasets cover a range of applications for multimodal activity recognition, none of them focused generically on activities of daily living in AAL environments. Moreover, none of them are provided simultaneously with data from videos, inertial units and ambient sensors. Our approach aims to alleviate this gap by providing a dataset captured in a heterogeneous, sensory rich environment comprised of a smart home system, a wearable sensor kit, and a domestic robot equipped with an RGBD camera.

## 3. Human Activity Recognition

Different algorithms can be suitable for the task of human activity recognition, depending on the nature of the data being addressed [68]. For RGB videos, although strategies based on classic feature extraction techniques still provide competitive results [69,70,71], Deep Learning (DL) architectures have led to increasingly accurate, state-of-the-art results, representing a very active field of research, as discussed by Zhang et al. [72]. Among the most influential studies on this subject is Simonyan and Zisserman [28], which presented the Tow-Stream ConvNets, characterised by a *spatial* and a *temporal* stream. The spatial stream consists of a Convolutional Neural Network (CNN) that classifies individual RGB frames from a video, while the temporal stream is a similar CNN which, instead of an individual image, processes a sequence of dense optical flow maps (horizontal and vertical), computed on a preprocessing step using a suitable algorithm [29,30,31], from a predefined number of frames. The scores obtained by both streams are then fused, in order to obtain a prediction. Most of the works found in literature built on the basic structure of the Two-Stream ConvNets, including the Temporal Segment Networks [18]. Recent literature has proposed different multiple-stream approaches that could include other input modalities [12]. Our work was based on the multiple stream paradigm, in which the temporal stream was extended to work with a combination of CNN and Long Short-Term Memory (LSTM), as proposed by Donahue et al. [32]. It is worth to notice that spatio-temporal approaches, usually based on 3D CNNs, have been a popular alternative to multiple-stream approaches such as ours [73,74,75]. In this paper, the approaches implemented for video classification consisted of combining multiple stream principles using optical flow maps, with feature extraction with a CNN and temporal modelling with LSTM.

With respect to depth videos, state-of-the-art results have been obtained from different approaches. Motion from depth images, including optical flow features computed over depth human silhouettes, along with features exracted from human joints, are usually employed to compose Hidden Markov Models (HMM) [76,77,78,79,80,81], or other representations such as Self-Organising Maps (SOM) [82]. The most successful approaches are based on features extracted from geometrical relationships on skeleton joints [83]. In the context of DL, some researchers investigated the introduction of preprocessing steps such as the computation of depth motions maps [84], or the computation of action maps from scene flow representations [19]. We did not include the depth videos as a modality for computing the temporal stream because there is not a direct correspondence between the preprocessing steps of the most successful approaches on this context and the algorithms that we have analysed for the other modalities. The three-dimensional version of the optical flow, the scene flow, could be computed based on RGB-D images [85], but led to poor results on exploratory experiments and, hence, were unconsidered. Nonetheless, we included the raw depth images as an additional condition for analysing the spatial stream.

Considering sensors other than video cameras, the survey by Wang et al. [86] defined four modalities: body-worn (i.e., wearable sensors such as smartphones or watches), object (i.e., sensors attached to objects, such as RFID or IMUs attached to utensils), ambient (i.e., sensors attached to to environment, such as door sensors or Bluetooth beacons), and hybrid (i.e., combinations of modalities, typical for smart environments). Here, we are interested on body-worn (specifically IMUs) and ambient sensors, which composed a hybrid setting for our experiments.

Regarding activity recognition based on IMUs, research has addressed scenarios that resemble devices that are expected to be actually worn by the users, such as smartphones and smartwatches [87]. Feature extraction methods include combinations between sequential minimal optimization (SMO) and Random Forest [25], statistical features feeding genetic algorithms [88], and Markov models [89]. DL architectures, such as Deep Neural Networks (DNN), Convolutional Neural Networks (CNN), autoencoders, Restricted Boltzmann Machines (RBM), and Recurrent Neural Networks (RNN) have also been successfully applied to this modality [33]. In this paper, we designed a module for inertial sensors that resembled the DeepConvLSTM by Rueda and Fink [90], in which a convolutional module would perform feature extraction and feed it to an LSTM layer.

Considering ambient sensors, approaches can be divided into two major categories; data driven and knowledge driven. Domain Knowledge based systems use ontology’s and semantic reasoning to aid in recognition. Chen et al. [91] and Liciotti et al. [92] used a knowledge driven approach, including a Partially Observable Markov Decision Process (POMDP) and exploited the task information, while the location is combined with the sensor events in the smart home. Data-driven is mainly focused on use of feature extraction, temporal clustering, and activity recognition. Medina-Quero et al. [93] proposed a method using fuzzy time windows (FTW) to segment the data set, followed by Long Short-Term Memory (LSTM) for activity recognition. Gochoo et al. [94] extracted fixed-length sliding windows into a sparse two-dimensional time matrix to use Convolutional Neural Networks (CNN) for activity recognition. Guo et al. [15] provided a data-driven framework for activity recognition from multiple residents using time clustering.

Although different possibilities for fusion of multimodal data using DL methods have been proposed, especially regarding to different inputs from multiple video streams [95], the most popular approach for dealing with heterogeneous data is to process each modality separately and fuse the obtained scores at a later stage [35,36], which we refer as *late fusion*. This was the approach adopted on all experiments performed in this paper. Considering neural networks, variation of this method that has been adopted is to fuse the outputs of the modules respective to each modality using a fully-connected layer [75]. Other approaches have also been proposed, such as the Correlational Recurrent Neural Network (CorrRNN) presented by Yang et al. [96].

## 4. Methods

The experiments were performed in order to evaluate the improvements that could be achieved by combining motion information from videos and inertial sensors with static, contextual information from ambient sensors at a smart home. The task was to classify high-level activities, possibly composed by complex sequences of actions, using time-localised data, which is certainly a requirement for a real-time decision-making system. In the literature review, summarised in Section 2, we did not find a dataset suitable for such analyses. Hence, we designed a data collection procedure and collected the HWU-USP dataset, presented in the next subsection. This dataset captures a set of daily activities performed in the simulated kitchen at the Robotic Assisted Living Testbed (RALT), part of the Edinburgh Centre for Robotics in Edinburgh [97]. It was recorded with ambient sensors such as switches installed on wardrobes and drawers, inertial sensors attached to the wrist of the dominant arm and to the waist of the participants, and videos recorded from the RGBD camera of a domestic robot placed in a fixed observing location.

Besides the construction of the HWU-USP database, we have also performed experiments with the UTD-MHAD [37], already mentioned on Section 2.4, one of the most widely used benchmarks for multimodal activity recognition from videos and IMU. This known dataset differs from ours on its granularity, with actions composed of short movements such as *clap*, all of them with approximately the same length of very few seconds. It also provides more homogeneous data, with the subjects cautiously positioned facing the camera, always in full face (on the HWU-USP dataset, images in profile and in full face are presented on different frames of the videos). Although this is suitable for work on gesture-based interfaces, it is realistic for daily activities such as the ones that we are interested in. Moreover, it is focused on motion information and does not provide data from ambient sensors, which limits our analyses. Another difference is that, whereas the HWU-USP dataset is provided with two inertial units placed on the subjects’ waist and dominant wrist, the UTD-MHAD provides inertial data from only one unit, worn on the subject’s right wrist. Nevertheless, it may provide an alternative benchmark for our evaluations, besides allowing comparisons with our dataset on the performance of successive predictions over time using the same classifiers. Those analyses will be better described on Section 4.3.

As for the classifiers, we built on DL architectures for data from video and inertial sensors, presented on our previous work [36]. The most relevant contributions of this paper are the models trained not only on data from those modalities, but also considering ambient sensors from the smart home. This data was pre-processed to compose tuples of structured, categorical data which could be introduced as an additional stream to be introduced on the top of the neural network originally implemented for classifying IMU data. The design of the resulting neural network will be depicted on Section 4.2.

### 4.1. The HWU-USP Activities Dataset

The multimodal datasets presented in Section 2 provide data from different kinds of videos and inertial sensors, but they did not include data from ambient sensors. The main contribution of our dataset is introducing the data from the smart home devices synchronously with videos and inertial sensors. Moreover, we provided videos of either activities made of repetitive patterns, such as *reading a newspaper*, and more complex activities with long-term dependencies, such as *preparing a sandwich*. This makes our dataset more realistic regarding the set activities, with respect to what could be expected on an actual AAL scenario, when compared to the others.

As already mentioned, the data collection was performed at the RALT laboratory [97]. The RALT is a 60 m${}^{2}$ (square meters), fully sensorised space designed to resemble a typical single level home comprising an open-plan living, dining and kitchen area and a bathroom and bedroom, and hosting a number of smart home, assistive technologies and domestic robots, such as the TIAGo robot, manufactured by Pal Robotics [98]. Besides collecting data from the smart home, people being recorded were asked to wear a wearable kit comprising of a smart watch and a sensor device to be installed on the belt, both equipped with IMUs. Furthermore, a Tiago robot was placed at a fixed location, to record data from its RGBD camera.

The data collection procedure received ethics approval from the Heriot-Watt University ethics committee on the 17th of November, 2019. A set of 16 volunteers participated on this study over the period of 2 weeks, performing a single repetition of each of the nine activities included in our protocol. This was to ensure to capture a degree of variability in the performance of each activity being recorded, including different timings for primitive actions and overall activities performed by different people. The participants, healthy volunteers with neither functional nor cognitive impairments, signed a consent form and the data collected did not include their identity (participants’ faces were also made anonymous by blurring the recorded image). Each participant was brought into the lab and the activities were explained and the participants were set up with the IMU’s in the kitchen. The following subsections will provide more information about the sensors and protocols used for data collection.

#### 4.1.1. Sensors and Modalities

The RALT is a ”Living-Lab” home-like environment designed to facilitate user-driven design and testing of innovative information and communications technologies (ICT) and robotic solutions for healthy ageing and independent living. In Figure 1, the whole environment is illustrated.

This environment is equipped with ambient sensors to perceive, monitor and understand occupancy’s daily activities. The sensors are positioned around the household with uniquely identified identity (ID), together with timestamp to indicate and record of occupancy’s activity. In our dataset, we recorded the sensors that were available in the kitchen and that would be meaningful for our purposes. Specifically, we considered four binary switches, two of which were positioned at the doors of two cupboards, respectively containing mugs and dishes, one at the door of the fridge, and one at a drawer used to store cutlery. We also considered the PIR sensor present in the kitchen, and the power measurements by the kettle, used for preparing tea.

The TIAGo robot is a mobile service robot designed to work in indoor environments. It has an extendable torso and a manipulator arm to grab tools and objects. Its sensor suite allows it to perform a wide range of perception, manipulation, and navigation tasks and is used for assisted living research in the RALT. For our data collection, we considered only data from its RGBD camera, an Orbbec Astra [99] device installed in its head. According to the manufacturer’s specifications, the range of this depth sensor lies within 0.6 and 8 meters. We positioned the robot in such a way that all activities and objects of interest were within this range. The colour VGA 640 × 480 at 25 fps and depth stream mode VGA 640 × 480 at 15 fps were used for the HWU-USP dataset. The TIAGo robot was placed in the environment with a clear view of the participants, at a fixed viewpoint across all recording sessions (see Figure 1).

As an wearable device for providing IMU measurements, we adopted the MetaMotionR, by MbientLab [100], a commercial device equipped with inertial, temperature, light and humidity sensors. The participants were asked to wear two MetaMotionR units, one of which placed at a wristband worn at the subject’s dominant arm, and the another placed at a clip worn at the subjects waist. These devices and placements are shown in Figure 2. We recorded data from the accelerometers and gyroscopes, synchronised using the robot’s internal clock. A sample from the dataset, considering the different modalities present in the dataset, is shown in Figure 3.

#### 4.1.2. Activities List

The activity list was based on the types of activities usually performed in kitchen environments. These were activities of daily living (ADL), tasks that require a level of functional capability and are completed in everyday independent living, such as cooking and cleaning. The activities also required that the participants manipulated a variety of objects and furniture in the kitchen, especially the cupboards, the simulated fridge and the drawer for cutlery, all of them equipped with binary switches. The participants were also asked to complete the list of activities in their own time. Intervals between recording each activity were implemented, such that the participants could look over the activity list and solve any doubts. The tasks were explained to the participants prior to completing the task, in which they were given a specific order and scripts to complete each of the tasks, such as the location of the items they were instructed to use, and relevant locations where they needed to carry out different actions. Since we were not recording sound, we gave instructions during the completion of the activities as well, so that the participants were not required to necessarily memorise all details respective to each activity. The data collection lasted approximately 20 min per participant as they completed the following set of activities, as illustrated in Figure 4. Notice that those activities have variable lengths, ranging from about 30 s to almost 2 min.

*Making a cup of tea*: taking the kettle to the sink filling the kettle, turning it on, collecting a mug and teabags from separate cupboards before combining and filling with water.*Making a sandwich*: collecting of a plate, bread, ham and cheese from the respective cupboards and fridge, and assembling with all the ingredients on the worktop.*Making a bowl of cereals*: collecting of the spoon at the cutlery drawer, the bowl and the cereal from separate cupboards, and the milk and honey from the simulated fridge, placing everything on the worktop and assembling.*Setting the table*: moving the prepared sandwich, tea and cereal from the worktop to the place mat on the kitchen table.*Using a laptop*: using a laptop while sat at the kitchen table, complicated with the cluttered environment from the previous activities.*Using a phone*: similar to “using a laptop”, but with a phone device instead, with both the laptop and the meal at the table.*Reading a newspaper*: similar to “using a phone”, but with the participant reading the newspaper.*Cleaning the dishes*: taking the bowl of cereals to the waste bin, dispose it from its content using the spoon, then pretend to wash it in the sink using a sponge. Due to the position of the sink respective to the robot’s positioning, the participant partially obscures this activity.*Tidying the kitchen*: returning the items to the cupboards and moving throughout the kitchen environment.

Each activity was performed from the same starting point to avoid classification due to starting configuration alone. The participants would walk into the kitchen environment and begin the activity. Once the activity was completed then the recording of the data was stopped. The starting positions of the objects in the smart kitchen environment was kept constant through the course of the data collection, to ensure consistency of the dataset.

The statistics regarding the lengths of the recordings, for each activity considered, are shown in Figure 5. The resulting dataset was composed of a total of 144 instances (i.e., 16 subjects performing a set of nine activities), which summed about 116 min. As shown in the figure, the average length of of the activities was around 48 s, which is considerably larger than the recordings of most other datasets (see Section 2). An important observation is that the proposed activities were designed at a high level of abstraction, so that most of them were composed by complex sequences of shorter-term actions. For example, to prepare a sandwich, the participant had to place a plate on the board, take the ingredients from the fridge, take the bread from the cupboard, assemble the sandwich, and so on. Based on the timestamps of the videos, fine-grained annotations may be provided as needed, so that each of those actions could be treated as a separate label. This could provide a different, more challenging scenario to be addressed on future research. In this paper, however, we are interested on the presentation of the data collection procedure, the dataset, and the multimodal framework for classification, which operates on high-level activities.

### 4.2. Classification Framework

For feature extraction and classification of activities recorded with videos and inertial data, we have proposed different DL architectures and compared the resulting models and their accuracies on a previous work [36]. Based on the results obtained in this previous paper, we chose the CNN and LSTM models as basis for our experiments. As it already mentioned, our main contribution was the introduction of contextual data from the ambient sensors of the smart home as an additional stream, specifically by including an additional input stream at the neural networks aimed at the inertial data. The different scenarios considered will be presented on the next subsections.

#### 4.2.1. Segment-Wise Classification and AAL Applications

Before presenting the methods proposed for multimodal activity recognition, it may be worth to discuss the type of applications that could benefit from either the HWU-USP activities dataset or the classification framework to be presented in Section 4.2. An AAL application that could be addressed is shown in Figure 6, which summarises scenarios proposed on related work [101]. The sensors made available on the data collection presented in Section 4.1.1, illustrated in the figure, provide inputs for the *activity recogniser* module, which is the focus of this paper. In an actual AAL environment, data would be gathered continuously from the available sensors, and predictions would be provided at each instant *t*. These predictions, referred in the figure as pred(t), consist of the outputs of the framework presented in Section 4.2, which will be evaluated and discussed in Section 5 and Section 6.

The next module that would be part of such an application would be the *behaviour scheduler*, a possible direction for future research. This module would be responsible for orchestrating the different ambient actuators and artificial agents (e.g., social robots or mobile applications), providing useful services or proactive behaviours for the inhabitants of the environment. For a real-time application, these behaviours are expected to be continuously adapted according to the predictions of the activity recogniser at each instant *t*.

State-of-the-art methods for multimodal activity recognition have been achieved remarkable results by processing previously segmented activities on its whole length [75]. Although this approach makes sense in the case of fine-grained activities, it would be of little use in contexts such as the scenario of Figure 6. There are two reasons for it. First, it requires that the activities have been previously segmented, which is not realistic for real-world applications. Second, it would require the activity to be finished before providing a reliable prediction, which could take more than a minute in the case of the activities of the HWU-USP dataset (see Figure 5). In this case, it is possible that the proactive behaviour of the AAL environment is no longer required, or does not make sense to be performed after the human activity is finished. For example, a robot may need to bring the user’s glasses while he is reading the news—it would make little sense to do so after the user has stopped this activity.

Therefore, we designed our framework so that the DL architectures process two-seconds-long segments, and the predictions over a longer sequence could be enhanced at decision-level, by averaging the output vectors at each segment. This is an approach commonly used for video-based activity recognition [28,35], and we extended it to the other modalities to provide a framework that is able to work with partial data.

#### 4.2.2. Data Preprocessing

Regarding the video modalities, following the proposal by Simonyan and Zisserman [28], we considered multiple streams. Videos were resized to 320×240 before any other preprocessing step. For data augmentation, we implemented random cropping for training, and cropping of all corners and the centre for testing, resulting in frames of size 224×224.

The spatial stream could be composed by individual RGB frames obtained from the videos, as on the original framework. We also adopted a similar approach for taking the depth frames as inputs. To do so, the depth frames had been converted to 3-channel, 8-bit RGB inputs with the same intensity on all channels, composing grayscale samples which could be employed in transfer learning scenarios.

Two approaches were considered for the temporal stream. The first was to feed the learning architectures with pairs of dense optical flow maps, as in the original two-stream ConvNets [28]. Those maps were generated with OpenCV implementation of the TVL1 algorithm [31] on each pair of successive frames on the RGB videos previously converted to grayscale images. The outputs of those algorithms consist of the horizontal and vertical estimations of the displacements of each point from one frame to another, assuming their intensities are preserved on both images. In the case of dense optical flow, all pixels on the image might be considered.

In relation to the inertial and ambient sensors, the recordings were made asynchronously. The alignment was performed independently for each of the 144 recording sessions of the dataset, so far referenced as *instances*. Regarding the inertial sensors, consider that, for a given instance, there is a set of $P_{sk}$ rows of data from a sensor $s_{k}$, k∈{1,2}, with $s_{1}$ being the inertial unit of fixed to the user’s waist, $s_{2}$ the inertial unit fixed to the user’s wrist (more sensors could be added to this framework, as needed). Let $x_{sk}\left(p\right)$ be a vector correspondent to the *p*-th row of data registered by sensor $s_{k}$, correspondent to a timestamp $t_{sk}\left(p\right)$ obtained from a global clock during the data collection procedure. The alignment procedure intends to obtain an aligned file composed by *Q* rows, equally sampled at a desired sampling rate *r*, starting from the highest timestamp registered by any of the sensors. The vector y(q) is the *q*-th row of data aligned from both sensors (i.e., the output data). The timestamp correspondent to this row of data is $t_{y}\left(q\right)$, computed as in Equation (Equation 1). For each index *q*, the method consisted of composing a concatenated aligned row y(q), composed of data from both sensors, by appending the tuple of data $x_{sk}\left(i\right)$, i∈[1,P], from each sensor $s_{k}$, so that $t_{sk}\left(i\right)$ is the lowest value among the *P* rows in the instance that satisfies $t_{sk}\left(p\right)>t_{y}\left(q\right)$.
(1)$$\left\{\begin{array}{c} t_{y}\left(0\right)=\max\left\{t_{s1}\left(0\right),t_{s2}\left(0\right)\right\}\\ t_{y}\left(q\right)=t_{y}\left(0\right)+q\cdot \left(1/r\right)\;,\;q=1,2,\cdots ,Q \end{array}\right.$$

For preprocessing the smart home data, the same alignment procedure was used to provide one tuple for each timestamp, allowing a one-to-one correspondence with each tuple of the inertial data. Apart from implementation details, this is equivalent to including a sensor $s_{3}$ to the above-mentioned alignment procedure, correspondent to the set of ambient sensors from the smart home. An inertial input to the DL architecture would consist of a sub-sequence of a recording session of length $N_{t}^{\mathrm{raw}}$. The data from the ambient sensors were introduced to an additional preprocessing step before feeding the DL models: the attributes of the $N_{t}^{\mathrm{raw}}$ correspondent samples on the ambient sensors were averaged, composing a feature vector. Finally, the data from inertial sensors and from the smart home ambient sensors were L2-normalised, in order to compensate the scales of each variable.

All experiments performed were based on classifiers being applied to two-seconds long data segments, regardless of the modality. One example of input, with all the different modalities represented on the HWU-USP dataset, is shown in Figure 7. For the UTD-MHAD, although data from ambient sensors is absent, the remaining modalities were arranged on the same structure.

#### 4.2.3. Deep Learning Architectures

Our multimodal strategy for video and inertial data was based on late fusion of the output vectors of each single-modality model. This approach is also known as decision-level fusion [75]. Indeed, we implemented independent neural networks for videos and IMUs, and, at a later stage, performed weighted averaging of the scores at the outputs of the softmax layers of each network. Data from ambient sensors of the smart home were introduced as an additional input vector on the same neural network used for classifying the IMU data, so that the resulting output vector could also be combined to the video output to provide a classification framework with all modalities considered.

In Table 5, the DL architectures employed for each modality are summarised. We began our analyses by considering two baseline architectures resembling the spatial stream of Simonyan and Zisserman [28]. A consolidated CNN model, the InceptionV3 [102], was employed to train two models for each dataset: one for processing RGB frames, and another for depth frames. We named these modalities *RGB* and *Depth*, respectively. The models were pre-trained on the ImageNet dataset [103], and had all their layers fine-tuned for training the activity recognition datasets under analysis.

For the video-based temporal stream, which processes optical flow maps, we implemented the neural network of Figure 8a. This consisted of a CNN, which was trained and evaluated previously for performing the same classification task. Their inputs were a set of flow maps respective to a single pair of frames. This input consisted of two-channels, which comprises the optical flow previously computed. The InceptionV3 architecture was adopted for this aim. We refer to the classification models composed by this CNN alone, without temporal modelling of a sequence of frames, as *optical flow (single frame)*.

To model sequences of optical flow pairs (i.e., the *optical flow (sequence)* condition), the 2048 features, extracted right before the softmax layer of the InceptionV3 architecture, were considered. The CNN was applied $N_{t}^{\mathrm{video}}$ times, each to the optical flow input respective to one timestep of a sequence of timesteps, generating a set of $N_{t}^{\mathrm{video}}$ feature vectors. Those were fed to an LSTM module, whose outputs were the input of a softmax layer for classification. The LSTM module was composed of 128 units and dropout of 50%, with sigmoid activation. L2-normalisation was employed for regularisation.

The same structure was designed for the IMU data, as shown in Figure 8b, with the difference that, instead of an InceptionV3, we used a 1D CNN for feature extraction, which performed convolution and pooling operations on the time domain. Let the length of these sequences be $N_{t}^{\mathrm{raw}}$. We implemented this CNN with three convolutional layers with kernel size 11 and ReLU activations, interspersed with max-pooling layers of kernel size 2. The convolutional layers were composed by 128, 256 and 378 units, respectively. Batch normalisation was introduced before the first and the last convolutional layers. This convolutional block, referred as 1D CNN in Figure 8b, was followed by an LSTM layer with 128 units, ReLU activation, and dropout of 50%.

Regarding the machine learning aspect, the most noticeable novelty in this work was the introduction of data from ambient sensors of the smart home on the learning framework, which could be done with the new HWU-USP dataset, but not with the UTD-MHAD. An additional input vector, composed by structured data from binary sensors and voltage measurements from the kettle, was added to the same network designed to learn features and classify the IMU data. One condition was included to process this input vector with a shallow neural network: a fully-connected neural network with two hidden layers composed of 512 and 256 units, respectively, with ReLU activation and dropout of 50% after each layer, and a softmax redout layer.

A feature-level fusion architecture between the IMU and ambient data was also proposed. The approach, as shown in Figure 8c, was to process the input of the ambient sensors in parallel to the convolutional and recurrent layers of the IMU architecture (Figure 8b), this time using a single fully-connected layer with ReLU activation function and dropout of 50%. The outputs of this layer would be concatenated to the features learnt by the convolutional module of the inertial data, and then classified with a softmax layer.

Skeleton joints, which may be extracted by RGBD cameras, were not considered on our framework. When designing our dataset, we were interested in providing a framework based on DL techniques, which have shown to provide good results for video classification on highly unstructured scenarios, closer to real-world applications. However, the proposed architectures could not be employed directly to data from skeleton joints without a feature extraction stage. To properly consider data from skeleton joints for our dataset, we would have to process it with unrelated techniques, which we understand to be out of the scope of this work.

### 4.3. Experimental Setup

For the implementations of the models presented in the previous subsection, we adopted the TensorFlow library, particularly the Keras module, which provides support for GPU training and evaluation. The models were trained on different hardware devices: the cluster Euler, at the Centre for Mathematical Sciences Applied to Industry (CeMEAI) at ICMC-USP, with GPU nodes provided with a Nvidia Tesla P100; a research computer at the Robots Learning Laboratory (LAR), at ICMC-USP, provided with a Nvidia Titan V GPU; and an ASUS TUF Gaming laptop, provided with a Nvidia Geforce RTX2060 GPU.

All architectures were fed with sequences which correspond to two-seconds-long segments of the recordings. For the case of the video modalities, the inputs were sequences of length $N_{t}^{\mathrm{video}}=15$ with period T=2 (i.e., the frames were downsampled on the temporal dimension to half of its original frequency) for the UTD-MHAD dataset, and $N_{t}^{\mathrm{video}}=25$ with T=1 for the HWU-USP dataset. For the inertial and ambient modalities, the length of the sequences were $N_{t}^{\mathrm{raw}}=100$ for both datasets, as the two of them were converted to a r=50 Hz sampling rate. The optimisation algorithm was Stochastic Gradient Descent (SGD) with learning rate $10^{-2}$, momentum 0.9 and decay $10^{-4}$. For the video models, training was performed for 40,000 steps, and, for the others, for 20,000 steps.

The evaluation protocol consisted of cross-subject training and testing, with a leave-one-out-approach. That means recordings from one subject were used for testing, while all others were used for training. Consequently, for each input modality, we have trained eight models, and reported the mean and standard deviations of their performance in the test sets. The predictions were obtained using the same principle as recommended in Simonyan and Zisserman [28]. They consisted of evaluating 25 segments on each session recorded, equally spaced between them, and the resulting scores of all outputs were averaged before to produce a prediction. This was done on all modalities. Although this could negatively affect our overall accuracy, this setting is more consistent to real-world applications in which an agent must take actions based on limited, time-localised information. Analyses of the confidence of the predictions through time could be performed, which allowed to better understand the behaviour of the classifiers on the activities of different levels of complexity of the HWU-USP-MHAD dataset, and to compare these results to those obtained with the simpler and shorter activities of the UTD-MHAD.

Fusion of the video streams and the other modalities was performed ad-hoc, after the predictions were already obtained and recorded. The procedure consisted of averaging the outputs of the modalities that were being combined, with different weights for different modalities. The accuracies on different multimodal scenarios were computed on the output vectors respective to this average. For HWU-USP dataset, the weights were set to 1 and 6 for the IMU and video modalities, respectively. For UTD-MHAD, they were set to 1 and 2. On both cases, the weights were chosen in order to maximise the accuracy obtained on each fusion approach.

## 5. Results

In this section, results of the experiments with respect to each modality are presented, along with multimodal approaches, in Table 6. The *RGB* and *Depth* modalities were computed by feeding an InceptionV3 newtwork with regular 3-channels frames extracted from the videos. In the case of the RGB frames, these consisted of the colour channels of the image, as usual in CNNs. In relation to the depth frames, the 16-bit inputs were converted to 8-bit maps, which were repeated on the three channels, composing grayscale images, already mentioned in Section 4.2.2.

The modality *optical flow (single frame)* refers to a modification of InceptionV3 network to receive a 2-channels input, which was fed with one pair of dense optical flow (see Section 4.2.2), hence considering only one pair of timesteps on the sequence. On the other hand, the *optical flow (sequence)* models refer to LSTM modules processing the features extracted by CNNs for two-seconds long segments of the recordings (see Section 4.2.3 and Figure 8a). The *ambient (shallow)* modality refers to a shallow fully-connected neural network applied directly to the subsequence (see Section 4.2.3), whereas the *IMU* modality refers to the one-dimensional CNN-LSTM models applied to the data from inertial sensors, also computed on two-seconds long subsequences (Figure 8b), the *IMU + ambient* comprises one multimodal setting with both modalities combined within the neural network (Figure 8c). Finally, the *Optical flow + IMU* and *Optical flow + IMU + ambient* multimodal conditions refer to the late fusion approach presented in Section 4.3, consisted of combining the output vectors of each modality before making a final prediction.

It is important to emphasise that all results were computed with predictions from the average output vector from 25 segments on the test data of each modality, and that the train and test partitioning followed a cross-subject approach with eight folds (see Section 4.3), hence the table shows the mean and standard deviation over these eight folds. In Table 6, we presented results on both the HWU-USP and UTD-MHAD datasets, despite the important differences existing between them (see Section 4).

The confusion matrices for part of the above-mentioned models, for both datasets, were computed in order to allow a more in-depth discussion on the behaviour of each model. For the HWU-USP dataet, these matrices are shown in Figure 9, and, for the UTD-MHAD, in Figure 10.

We have provided another analysis to evaluate how each model performs across the different segments used to compute the final prediction. These results may lead to important discussions when considering which approach will be adopted for a real-time application, in which partial results computed on a limited range of time might be used in decision-making systems. Figure 11 and Figure 12 present, respectively, for the HWU-USP and UTD-MHAD datasets, the maximum score on the output vectors correspondent to the actual class, across each of the 25 segments used for prediction. For example, taken an input that belongs to the *laptop* class, if the output vector of the first sequence fed to a given classifier gives a 25% confidence for predicting the correctly, and the first sequence of another instance from the same class gives a 32% confidence, the value considered for the figure will be 32%.

### Comparison with the State-of-the-Art

HWU-USP database is being presented for the first time in this paper, hence the above-mentioned results are the first to be ever published. For this reason, there is still no literature to compare it with. On the other hand, UTD-MHAD dataset is a widely used benchmark which we can use to evaluate our multimodal approach with only videos and IMU, since this dataset does not provide data from ambient sensors. In Table 7, a collection of results from the literature was put along with the best result that we achieved. To select those studies, we followed the criteria that videos and IMU data were both employed, preferably without skeleton data, so that the comparison with our approach would be as fair as possible. Besides, we only considered studies in which it was explicitly stated that the evaluation protocol was cross-subject. However, it is hold to note, that only Wei et al. [75] adopted a leave-one-out approach, similar to ours, while the others followed the protocol by Chen et al. [37], which used a hold-out approach with half of the data being used for testing.

## 6. Discussion

The RGB frames (i.e., the *spatial stream* by Simonyan and Zisserman [28]) led to accuracy measures slightly above random choice for both datasets (i.e., 6.39% for 27 classes on the UTD-MHAD, and 19.57% for 9 classes on the HWU-USP dataset). The depth frames did not led to better results for UTD-MHAD (i.e., 5.46%), but led to an important improvement for HWU-USP dataset (i.e., 36.36%). Still, both approaches led to poor results, if compared to the other models. These results differ from those obtained for video datasets in the literature of multiple stream classification methods [18,28], in which the spatial stream alone led to competitive performances.

Even though, the low accuracy obtained in our experiments was expected due to the nature of the datasets analysed. Applied directly to RGB, or even depth images, a CNN is able to distinguish between the objects, backgrounds and other appearance-based aspects within a scene. Thus, it may be effective when comparing videos from heterogeneous datasets with large inter-class variability regarding those aspects. This may lead to comparatively high accuracy even if motion information was disregarded. The datasets considered in this study present constant background and a limited variability regarding other appearance aspects. Different from the UTD-MHAD, the HWU-USP dataset was recorded from a perspective in which the subjects changed their position constantly with respect to the depth dimension, which may explain the improvements that happened only for this dataset when compared the depth to the spatial models. Nevertheless, for both datasets and any other that shares these characteristics possibly inherent to home environments, a reliable classification method might be based on motion information.

Motion information contained in dense optical flow maps (i.e., the *temporal stream*) led to expressive improvements, even on the *single frame* scenarios. These models were, by far, the ones that led to the highest accuracy on the HWU-USP dataset, which points to the relevance of motion information from computer vision on the scenario analysed. The results were also the bests for the UTD-MHAD, however the IMU condition was still competitive.

When comparing the *single frame* to the *sequence* optical flow architectures, HWU-USP dataset was characterised by a greater increase in accuracy (i.e., 86.72% to 93.75%) than the UTD-MHAD (i.e., 82.47% to 84.79%). This was probably because the HWU-USP dataset is composed of longer recordings with longer-time dependencies. This illustrates how the LSTM-based module is effective in modelling the long-term dependencies that were introduced.

Compared to the video modalities, especially the *optical flow (sequence)*, IMU-based models performed better on the UTD-MHAD, in which the 82.23% accuracy was even competitive when compared to the optical flow models, than in the HWU-USP dataset, which appeared to be favourable for computer-vision approaches. This was probably because the actions on the UTD-MHAD dataset are shorter and more well-defined, so that the most discriminative features were present on most snippets of the inertial data. For the HWU-USP dataset, some of the activities are complex, composed of sequences of actions that may, isolated, be part of different activities. The visual information contained on videos may be more informative than the IMU data with respect to these more challenging dependencies.

Analyses involving the binary data from the ambient sensors of the smart home could be made only on the HWU-USP dataset, and led to promising results. On its own, feeding a shallow fully-connected network with minimum preprocessing, this modality led to an accuracy of 51.39%, which is expressively below the 65.56% obtained by the one-dimensional CNN-LSTM applied to the IMU data alone. However, when combined, the model hit the accuracy of 74.30%, the best performance obtained without the use of optical flow data.

When combined, the models in which the optical flow models were fused to the other modalities led to the best accuracies. For the UTD-MHAD, this approach led to 92.33%. For the HWU-USP dataset, two conditions were considered: combining the optical flow and the IMU models, as with the UTD-MHAD, and combining the optical flow model to the IMU + ambient model (see neural architecture on Figure 8c). For the first condition, the accuracy was 96.53%, an increment of almost three percent points when compared to the optical flow model on its own. For the second condition, which was possible only because we had made available data from the smart home sensors on the HWU-USP dataset, the accuracy was 98.61%, which may be seen as a remarkable result.

It may be worth discussing some aspects regarding the confusion matrices shown in Figure 9 and Figure 10. Considering the models for the HWU-USP dataset, the most solid observation is that the *cereals* and *tidy* activities are the most sources of wrong predictions on the models with higher accuracy, for either the computer-vision or IMU models. The introduction of the ambient sensors caused an important impact on the recognition of these classes, bringing the error down to zero, which may explain its relevance of the accuracy results of Table 6. Multimodal models provided basically a reduction on the mistakes made in some classes, when compared to the predictions made by each single-modality model. The UTD-MHAD models performed more uniformly across the different modalities, which may explain why the results did not vary too much for the single-modality approaches. For the multimodal scenario, the classes with less precision on each modality seem to have been compensated, causing the observed increment of accuracy.

The confidence scores through time, shown in Figure 11 and Figure 12, seem to have been expressively improved on the UTD-MHAD when comparing the single frame to the sequence approaches. However, these accuracy improvements were more prominent on the HWU-USP dataset, even though the differences of the confidence scores through time seemed to be smaller on these figures. This was because the activities from the UTD-MHAD dataset were all short and made of simple gestures, hence a small snippet on the middle of a video recording could be more informative of the actual activity, providing a correct prediction with high confidence. The same is usually not true for the HWU-USP dataset.

For the UTD-MHAD, the confidence over segments (Figure 12) also presented important differences between the conditions. The *optical flow (sequence)* model provided high confidence scores on the segments closer to the middle of the recordings, in which it differs from the *optical flow (sequence)*, with confidence scores approximately uniform on the whole sequences. For several classes, the IMU model provided high confidence scores only for the first half segments of each modality. The multimodal *IMU + optical flow* provided an improved version of the *optical flow (sequence)* model, except for activity 22, which appears to have its confidence degraded by the IMU scores.

On the other hand, for the HWU-USP activities dataset (Figure 11), a diverse behaviour of the classifiers on each modality was observed. Regarding the optical flow conditions, the *sequence* approach is less uniform across segments than the *single frame*, but seems to provide higher confidence on certain parts of the activities. The activities performed with participants sitting down and performing repetitive movements (i.e., *laptop*, *smartphone* and *newspaper*) led to higher confidence scores for the IMU modality. The combination between IMU and ambient sensors increased drastically the confidence of the *cereals* class across all segments. The *tea* activity led to high confidence scores on its ending, especially when considering the *optical flow (sequence)* model. An increment on this same region may also be seen when combining the ambient sensors to the IMU, which may be due to the power measurements, which change only during the final moments of the *tea* activity, when the participant turns on the kettle. Few differences may be seen when comparing the *IMU + optical flow* and the *IMU + smart home + optical flow* conditions, however the final segments of the *cereals* activity seems to reflect the most noticeable increment.

By analysing the confusion matrices in Figure 9 and the heat maps in Figure 11, it becomes clear that the predictions on the *cereals* class were the most benefited, which explained the expressive improvements aggregated to the *IMU + ambient* and the *optical flow + IMU + ambient* models with respect to the conditions without this modality. The increment of the confidence on the last segments of the *tea* activity (Figure 11) is also worth a mention, since it was probably due to the power measurements of the kettle, which was turned on at the end of all recording sessions of this activity. In any case, the improvements provided by the multimodal models give additional confirmation on the usefulness of combining videos and IMU modalities whenever they are both available, corroborating to other results from related work [35,75,106].

Comparisons with the state-of-the-art datasets, made only with the UTD-MHAD, pointed that, with respect to videos and inertial sensors, our methods led to results that were compatible. Our best approach led to an accuracy of 92.33%. The reference results, 79.10% for the multimodal condition, provided on the presentation paper of the UTD-MHAD dataset, were successively surpassed on the following years. Models that consider skeleton data led to the higher accuracies, such as in El Din El Madany et al. [104], which hit 93.26%. Nonetheless, these results are not comparable to ours, since this approach considered information extracted from skeleton joints, an additional, rich data input.

Different approaches restricted to the IMU and the video data have also been proposed. Imran and Raman [105] performed experiments with different sets of modalities, and hit 92.32% using RGB videos and inertial sensors, a result which is very close to ours. The most successful approach restricted to these modalities, nonetheless, was provided by Wei et al. [75], which hit 95.60%. This was the only result on the literature that surpassed ours without the use of skeleton information. We have included the RGB-only and inertial-only results in order to situate their results with respect to ours. It is important to note that the video-based method presented by them actually performed less accurate than ours (76.0%, against 84.8% of our approach), which means that the overall accuracy on their work benefited especially from the inertial-only model, which hit 90.3%, against 82.2% of ours.

However, the IMU architecture was based on a two-dimensional matrix representation of the input data, which required the whole sequence to provide a representation. This would not allow a segment-wise classification such as the approach proposed by us, in which the neural network processed two-seconds long data segments, and therefore its application would not be possible in scenarios with partial data, such as the real-time decision-making systems that would be expected on AAL environments.

## 7. Conclusions

In this paper, we presented the HWU-USP activities dataset, collected at the RALT lab in Edinburgh at Heriot-Watt University. More specifically, the dataset was composed of RGB and depth videos from the camera of a TIAGo robot, data from IMU sensors attached to the users’ wrist and waist, and a set of ambient sensors (i.e., switches at the doors of wardrobes and drawers, motion sensors and power measurements) from a smart home. The objective was to build and study a multimodal dataset composed of RGB and depth videos, inertial and ambient sensors from a smart home in the context of activities of daily living, all of them sharing a kitchen environment and performed in the context of a regular breakfast. A set of 16 participants performed 9 activities, resulting in a total of 144 instances that composed 116 min of recordings in total. All data were stored, made anonymous and will be available to the research community.

This dataset allowed the proposal of multimodal approaches involving not only videos and data from inertial sensors, but also ambient sensors. To the best of our knowledge, this is the first public multimodal activities dataset that provides these three modalities altogether and synchronously. We also proposed a deep learning framework to perform experiments on a multimodal approach. It is based on two-dimensional CNN modules for feature extraction on RGB frames, depth images and optical flow pairs, and LSTM layers for temporal modelling, when applicable. Data from inertial sensors were fed to a similar architecture, with a one-dimensional CNN being applied to extract features to be modelled by a LSTM module. For these modalities, we performed the same experiments on both the HWU-USP and the UTD-MHAD datasets. Results varied from one modality to another, especially for the HWU-USP, in which the architectures based on computer vision, specifically after computing dense optical flow, performed significantly better. These differences were smaller for the UTD-MHAD dataset.

The data from the ambient sensors, present only on the new HWU-USP, were introduced as an additional channel of information on the neural network that processed the inertial data, with no feature extraction: the binary variables were fed to a fully-connected layer whose output was concatenated to the IMU features extracted by the CNN-LSTM modules. The presentation of this fusion architecture is another contribution of our work As expected, the introduction of this modality led to expressive improvements in accuracy. The best multimodal model led to a very high accuracy, which points to the relevance of considering different sources of data to perform activity recognition tasks.

Future work will apply the models trained with this dataset to experiments in the smart home, allowing interventions to be made based on the predictions provided. This may be promising for application scenarios involving human–robot interaction (HRI). For example, a robot may use the successive predictions of an activity recognition framework to decide whether it might remember an user to take his medicines when he is having a meal, or bring an used glass from the living room to the kitchen when the user is washing the dishes. This may be important for designing Ambient Assisted Living solutions with automated technologies, such as robot carers, for monitoring the inhabitants of a smart environment. Moreover, fine-grained annotations may be provided for training models that suit most of those application scenarios and in accordance with the necessities that may arise during those experiments.

To react proactively to the users’ needs, this type of applications requires a framework able to provide reliable predictions in real-time, before the user finishes his current activity. This requirement was addressed by our approach, which relied on two-seconds-long segments, whose predictions may be combined to provide better results. In this sense, another direction for future research is to analyse how these predictions may be employed in real-world scenarios in order to complete the most adequate proactive behaviours in a timely manner. We provided an analysis of the confidence of the predictions across segments, which may give a hint on the performance of the framework in real-time applications. For other applications, in which those requirements are absent, experiments with longer segment lengths may be designed, which may foster research on novel learning architectures.

Although the classification methods provided excellent results when the video modality was present, there is still room for improvements regarding the other modalities. Such developments are important because, for real AAL environments, the video data may be frequently unavailable. This may be due to privacy issues or technical limitations, for example if the videos can only be registered by the camera of a social robot, which may not always be accompanying all the inhabitants of the environment. Yet, the very high accuracies provided by the video methods may serve to provide labels on a semi-supervised scenario for new data collections, which may rely on more cameras and more visual perspectives.

## Figures and Tables

**Figure 1 sensors-21-00768-f001:**
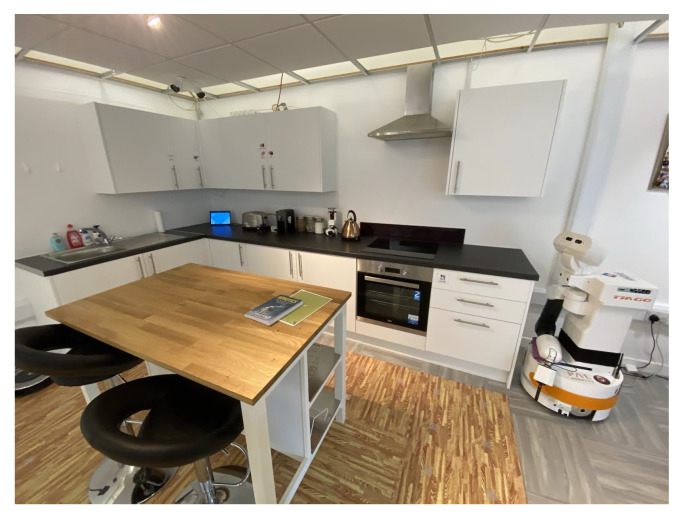
Environment in which the data collection was performed, with the TIAGo robot positioned on the corner of the kitchen (on the right side) during the recording sessions.

**Figure 2 sensors-21-00768-f002:**
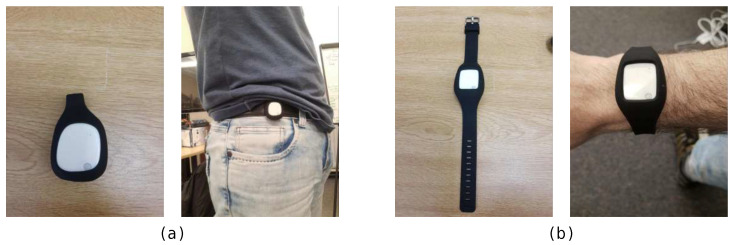
Inertial sensors attached to (**a**) a waist clip; and (**b**) a wrist band.

**Figure 3 sensors-21-00768-f003:**
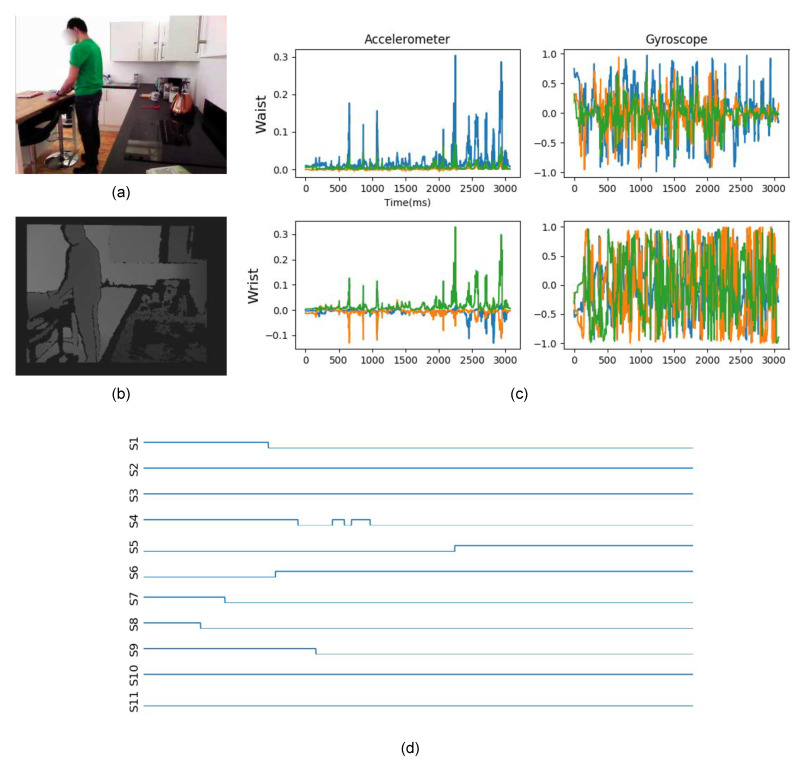
Sample of the dataset collected, consisted of (**a**) an RGB and (**b**) a depth image, both related to one timestep; (**c**) raw data from the inertial sensors, related to a whole sequence; (**d**) raw data from the ambient sensors (binary), where $S_{k}$ correspond to one of the *k* sensors available.

**Figure 4 sensors-21-00768-f004:**
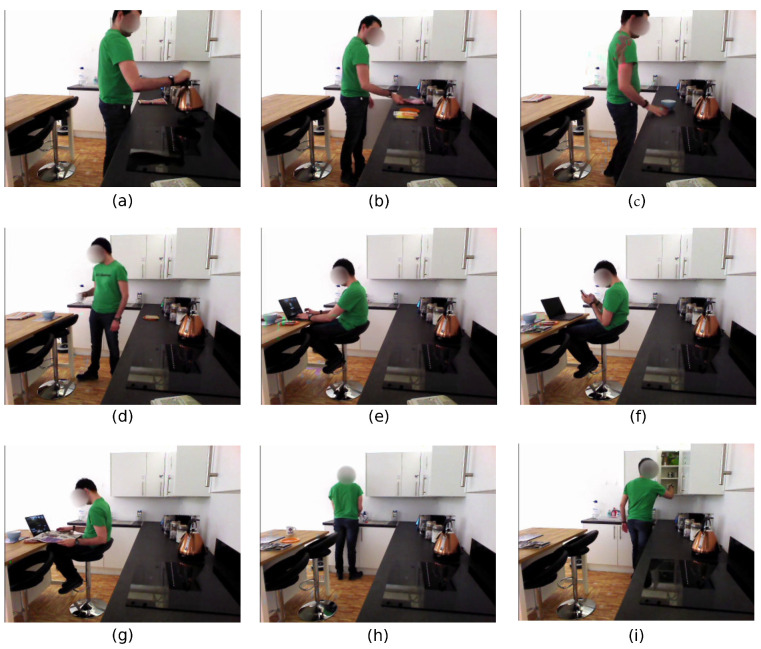
Sample frames of the activities considered for the dataset. (**a**) making a cup of tea; (**b**) preparing a sandwich; (**c**) preparing a bowl with cereals; (**d**) setting up the table; (**e**) using a laptop; (**f**) manipulating the cell phone; (**g**) reading a newspaper; (**h**) washing the dishes; (**i**) cleaning the kitchen.

**Figure 5 sensors-21-00768-f005:**
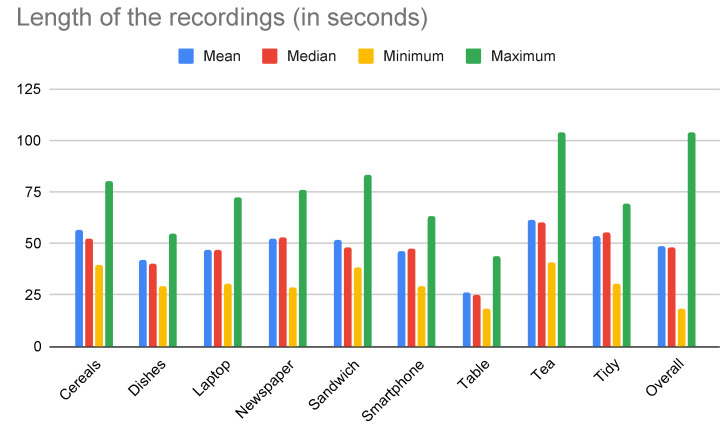
Statistics regarding the lengths of the recordings for each of the activities in the dataset.

**Figure 6 sensors-21-00768-f006:**
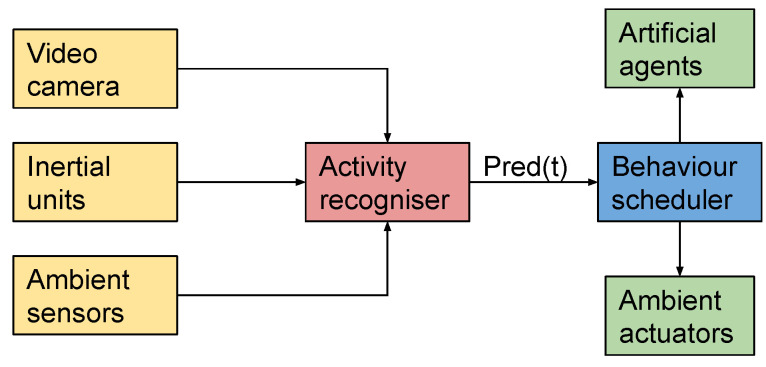
Example of an AAL scenario expected to be addressed by the proposed framework.

**Figure 7 sensors-21-00768-f007:**
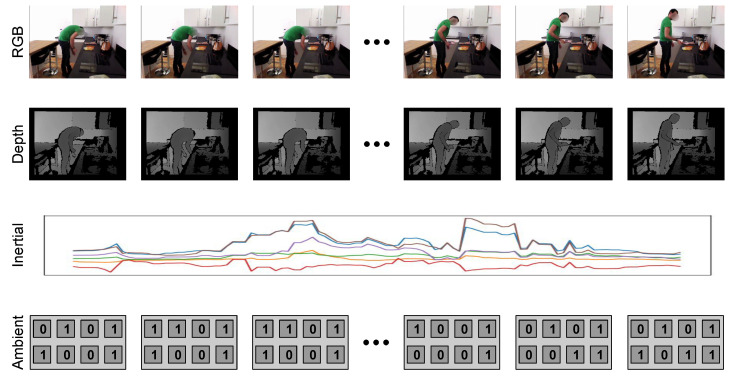
Example of two-seconds long segment fed to the architectures that process each modality.

**Figure 8 sensors-21-00768-f008:**
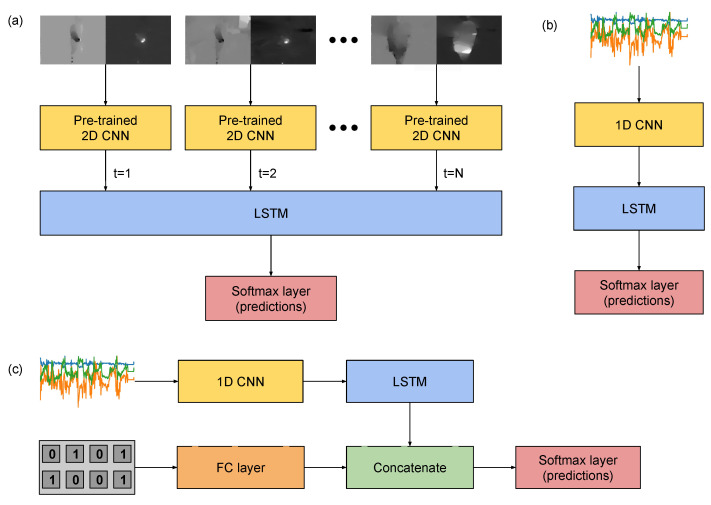
DL Architectures considered for each input modality. (**a**) Video inputs preprocessed by optical flow (i.e., two-channels input maps) algorithms. (**b**) IMU inputs, with a custom one-dimensional CNN as a feature extraction step before feeding a LSTM layer. (**c**) Multimodal scenario with fusion between inertial and ambient sensors within the modules of the neural network.

**Figure 9 sensors-21-00768-f009:**
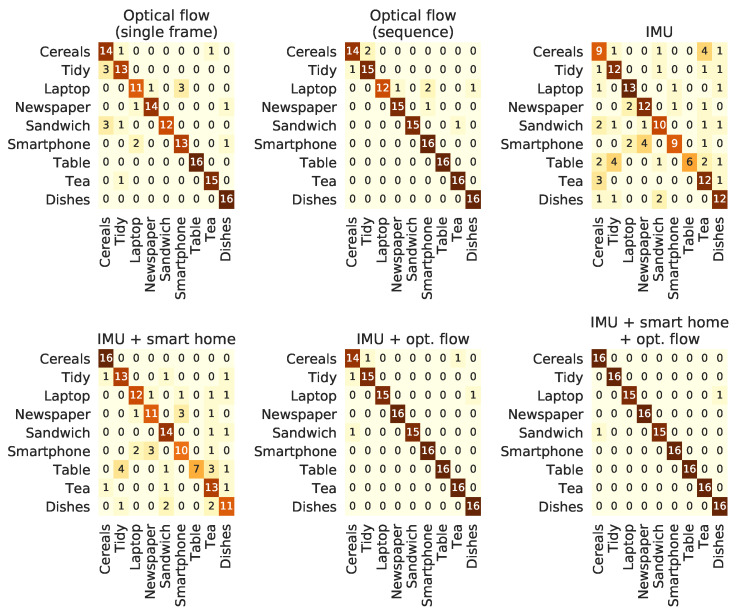
Confusion matrices of different input modalities and architectures for classifying the HWU-USP dataset. The values consist of the summed number of predictions over all folds.

**Figure 10 sensors-21-00768-f010:**
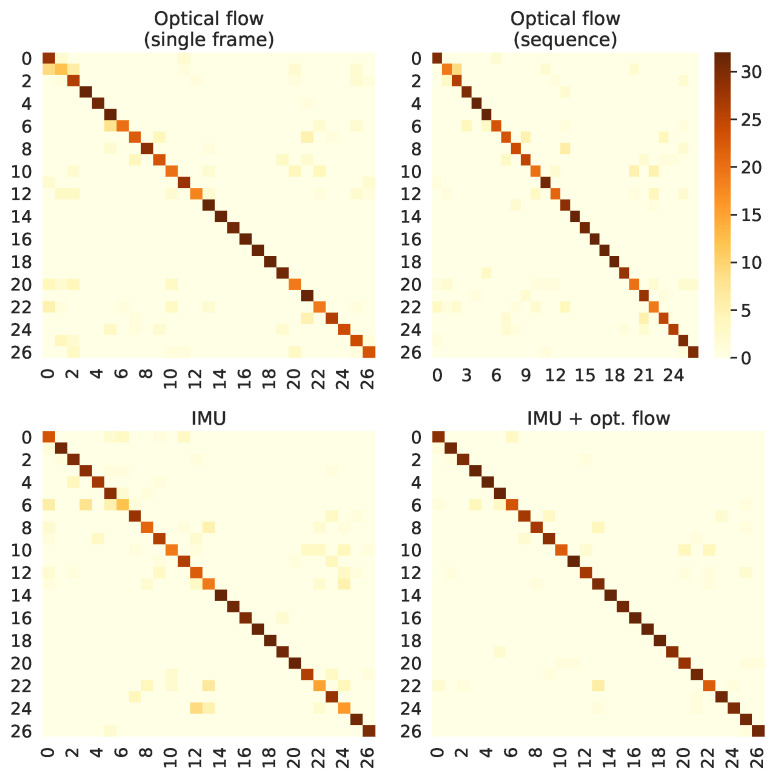
Confusion matrices of different input modalities and architectures for classifying the UTD-MHAD dataset. The values consist of the summed number of predictions over all folds.

**Figure 11 sensors-21-00768-f011:**
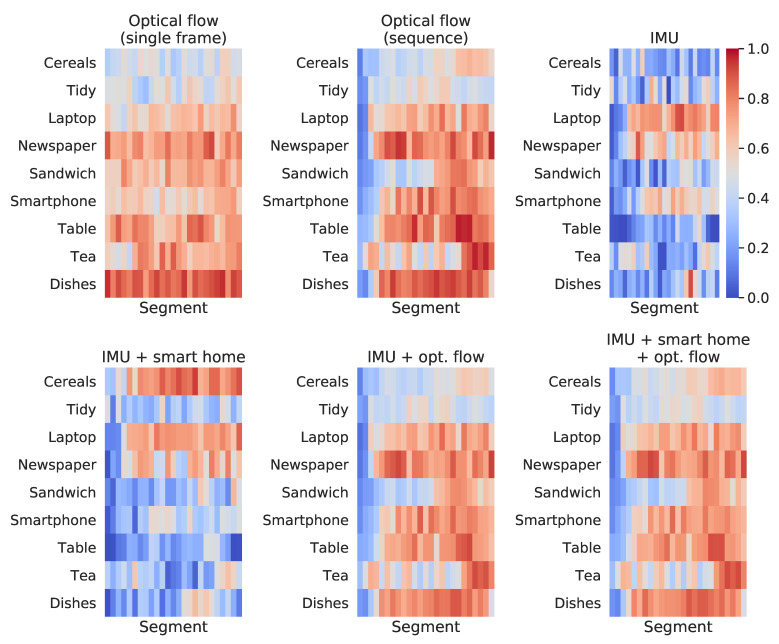
Confidence of predicting the correct label for the HWU-USP dataset, at each of the 25 segments evaluated, equally spaced between them.

**Figure 12 sensors-21-00768-f012:**
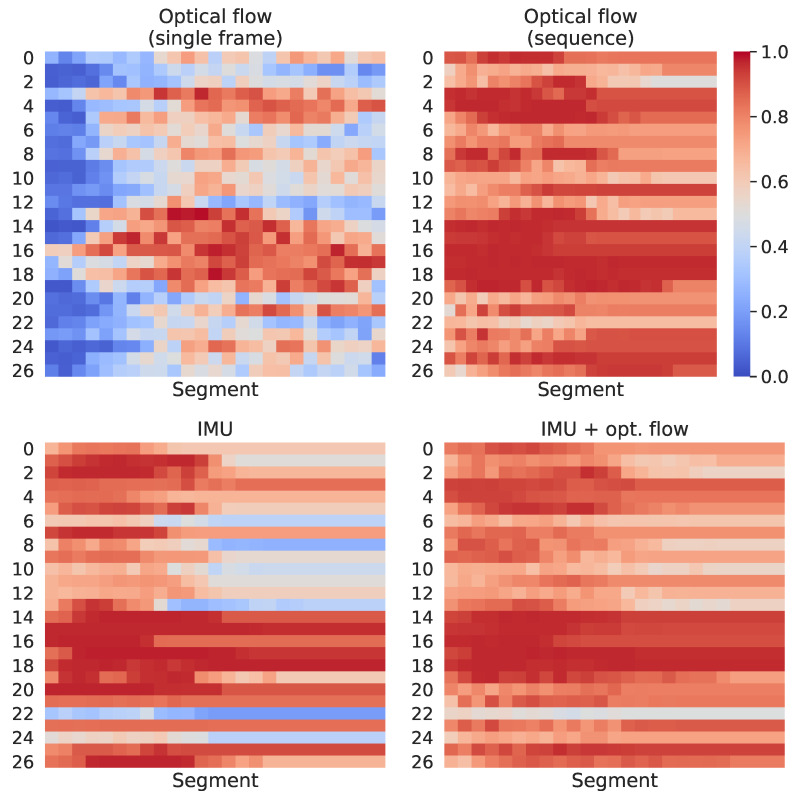
Confidence of predicting the correct label fot the UTD-MHAD, at each of the 25 segments evaluated, equally spaced between them.

**Table 1 sensors-21-00768-t001:** Video datasets made available and widely used in related works.

Dataset	Number of Instances	Categories	Source	Pre-Segmented
UCF101 [16]	13,320	101	YouTube	Yes
HMDB51 [38]	6766	51	Movies, YouTube, etc.	Yes
CCV [39]	9317	20	YouTube	Yes
Hollywood2 [40]	1707	12	Movies	Yes
Sports-1M [41]	+1 M	487	YouTube	Yes
Kinetics 700 [42]	+600 K	700	Youtube	Yes
THUMOS [43]	+23,700	101	YouTube	No
ActivityNet [44]	13,837	203	YouTube	No

**Table 2 sensors-21-00768-t002:** Selection of datasets for depth videos, adapted from the list by Amir Shahroudy et al. [46].

Dataset	Classes	Subjects	Repetitions	Instances
ORGBD [47]	7	24	2	336
MSR-DailyActivity3D [48]	16	10	2	320
3D Action Pairs [49]	12	10	3	360
RGBD HuDaAct [50]	13	30	-	1189
NTU RGB+D 120 [51]	120	106	-	114,480

**Table 3 sensors-21-00768-t003:** Datasets based on environmental or wearable sensors. Except for the OPPORTUNITY dataset, the IMUs were all contained on wearable devices.

Dataset	Sensors	Rates	Attributes	Subjects	Classes
OPPORTUNITY [52]	Wearable accelerometers: 12 Wearable IMUs: 7 Wearable tags: 4 Objects’ accelerometers: 12 Objects’ gyroscopes: 12 Environmental accelerometers: 8 Switches: 13	64 Hz 30 Hz 87 Hz 64 Hz 64 Hz 98 Hz 100 Hz	242	4	17
PAMAP2 [53]	Colibri wireless IMUs: 3 Heart monitor: 1	100 Hz 9 Hz	52	9	18
REALDISP [54]	Xsens IMUs: 9	50 Hz	120	17	33
SBHAR [17]	Samsung Galaxy S2 IMU	50 Hz	561	30	12
Skoda [55]	IMUs: 20	98 Hz	141	1	10
DG [56]	IMUs: 3	64 Hz	9	10	2

**Table 4 sensors-21-00768-t004:** Multimodal datasets, provided with videos, IMU sensors, and possibly others.

Dataset	Sensors	Rate	Subjects	Classes	Instances
CMU-MMAC [60]	Cameras: 5 Microphones: 5 Wired IMUs: 5 Wireless IMUs: 4 Motion capture: 1 eWatch (accelerometer)	30 Hz or 60 Hz - 120 Hz 60 Hz 120 Hz -	18	5	90
Berkeley-MHAD [61]	Motion capture: 8 Stereo cameras: 2 Quad cameras: 2 Microsoft Kinect: 2 Shimmer IMUs: 6 Microphones: 4	480 Hz 22 Hz 22 Hz 30 Hz 30 Hz 48k Hz	12	11	660
UTD-MHAD [37]	Microsoft Kinect: 1 IMU: 2	30 Hz 50 Hz	8	27	861
C-MHAD [62]	Webcam: 1 Shimmer3 IMU: 2	15Hz 50Hz	12	12	240
50 Salads [63]	Microsoft Kinect: 1 Accelerometers: 11	30 Hz 50 Hz	25	51	966
JIGSAWS [64]	da Vinci (kinematic data): 1 Stereo camera: 1	30 Hz 30 Hz	8	15	103
ChAirGest [65]	Microsoft Kinect: 1 Xsens IMUs: 4	30 Hz 50 Hz	10	10	1200
UR Fall Detection [66]	Microsoft Kinect: 1 x-IMU: 1	30 Hz 256 Hz	5	5	70
TST Fall Detection V2 [67]	Microsoft Kinect: 1 Shimmer IMUs: 2	30 Hz 50 Hz	11	8	264

**Table 5 sensors-21-00768-t005:** Summary of the DL architectures employed for each architecture.

Model	Input Description	Structure
Spatial	RGB frame	InceptionV3 [102]
Depth	Grayscale frame	InceptionV3
Optical flow (single frame)	Optical flow pair	InceptionV3 with two input channels
Optical flow (sequence)	Sequence of optical flow pairs	See Figure 8a
IMU	Sequence of raw data	See Figure 8b
Ambient (shallow)	Average vector from sequence of tuples	Fully-connected NN with two hidden layers
IMU+ambient	Two inputs (IMU and ambient)	See Figure 8c

**Table 6 sensors-21-00768-t006:** Accuracy Measures (%) for each input modalities, for UTD-MHAD and HWU-USP datasets. Models for a single input modality and multimodal models are listed. The accuracy shown is the mean value of 8 cross-subject folds (i.e., leave-one-out cross-subject evaluation protocol), with inputs from a single subject being left for testing, a costly, yet rigorous evaluation protocol.

	UTD-MHAD	HWU-USP
RGB (single frame)	6.39±2.16	19.57±6.76
Depth (single frame)	5.46±2.29	36.36±7.28
Optical flow (single frame)	82.47±5.42	86.72±6.74
Optical flow (sequence)	84.79±5.25	93.75±3.33
IMU	82.23±6.55	65.56±13.16
Ambient (shallow)	-	51.39±4.61
IMU + ambient	-	74.30±11.09
Optical flow + IMU	92.33±5.40	96.53±3.87
Optical flow + IMU + ambient	-	98.61±2.41

**Table 7 sensors-21-00768-t007:** Comparison between our model and others in the literature that deal with similar modalities, for the UTD-MHAD dataset. All of those evaluations adopted a cross-subject protocol.

Method	Modalities	Accuracy (%)
Chen et al. [37]	Depth + IMU	79.10
El Din El Madany et al. [104]	Depth + IMU + skeleton	93.26
Wei et al. [75]	RGB-only	76.00
Wei et al. [75]	Inertial-only	90.30
Wei et al. [75]	RGB + inertial	95.60
Imran and Raman [105]	RGB + inertial	92.32
Ours	RGB + inertial	92.33

## Data Availability

The data presented in this study are openly available in the Dryad Digital Repository, at https://doi.org/10.5061/dryad.v6wwpzgsj.

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
