# Peer review of "Activity Recognition for Ambient Assisted Living with Videos, Inertial Units and Ambient Sensors"

_sensors, 2021, doi:10.3390/s21030768_

Round 1

Reviewer 1 Report

This paper is related to the human activity recognition with videos. The introduction presents the motivation and aim of the study. The authors reviewed the different datasets, but some details must be added about the data acquisition in the different datasets, including devices used, frequency, and so on. The activity recognition methods must be tabulated to clearly view the results. The methods and results are clearly presented and discussed.

Author Response

The authors reviewed the different datasets, but some details must be added about the data acquisition in the different datasets, including devices used, frequency, and so on.

The datasets presented in Table 2 were all recorded with one version of the Microsoft Kinect, and a statement on this issue, with the sample frequencies allowed by those sensors, was introduced (see lines 197-199). In Table 3, the names of the devices were introduced for the datasets that used a commercial device and referenced it on the supporting material. The sample frequencies were obtained from the datasets and included to the table. We did the same for the datasets in Table 4. Regarding the inertial units, it may be worth to notice that some datasets relied on devices built by themselves, instead of employing commercial solutions.

The activity recognition methods must be tabulated to clearly view the results.

Table 5 was included for a more straightforward presentation of the details of each model. This table was provided with the same names and placed on the same order as Table 6, in order to allow a clearer visualisation of the results.

Reviewer 2 Report

In the paper, the authors build a new dataset which may support to recognize heterogeneous activities of daily living centers in home environments considering simultaneously data from videos, wearable IMUs and ambient sensors. Multimodal data can simplify model design and improve accuracy. In terms of privacy protection, the collection of data can be minimized while ensuring accuracy. It can be seen from the results that increasing the data of IMU and ambient sensors really helps a lot to improve the accuracy. And the authors propose a Deep Learning (DL) framework, which provides multimodal activity recognition based on either videos, inertial sensors and ambient sensors from the smart home.

As a data set, the code for loading data should be included, including the time alignment part. The biggest regret of the article is that the model itself is limited to a fixed segment of 2 seconds, but the data set is about one minute per segment.

The data set description of HWU/USP-MHAD lacks statistical information like Table 4, including average video length, minimum and maximum length. After I downloaded the dataset found that: Neither the number of action types nor the number of videos in this dataset exceeds that of other datasets. Is it only part of it or is it all already?

If you proposed a new data set, but immediately achieved very high accuracy (98.61%), that means this data set is not difficult enough. Consider adding some more complex actions, or combinations of actions as the challenging split of the data set.

There are some issues in the article that need to be improved:

  1. Is it possible to replace all “depth videos” with a unified expression, such as RGB-D videos?
  2. What is the main difference or contribution of the proposed dataset? Although in the paper, the difference is mentioned, what it can be driven or what it can be solved by this dataset. Please show the main contribution of this dataset.
  3. A new Deep Learning (DL) framework is designed, however, there is nothing about the structure of the new Deep Learning model.
  4. There is too much unnecessary description of other unrelated content that leads the article loss principal line. Please remove some unrelated content such as amount of description of other dataset.
  5. Please show more contribution of the proposed dataset, and provide a meaningful application, not just segmentation, classification. The proposed dataset needs to help, support, drive some application development.
  6. Line 491. Equation 1. The meaning of this symbol is not clear, consider redesigning this equation and explanation.
  7. Figure 5. The description of ambient is a bit rough, the value at each moment should be different.
  8. Figure 6. Lack of explanation for figure (c).
  9. Wrong number in Table 4, line ChAirGest, column size.
  10. Figure 3(d) It looks messy, consider removing these black borders or combining them into one chart.

The input of the model is limited to a fixed segment of 2 seconds. This is very detrimental to actual use and deployment. I hope that future tasks can consider using longer time series for classification and using an unfixed camera.

Author Response

As a data set, the code for loading data should be included, including the time alignment part. 

The code was included to the zip file at the repository, as well as the aligned versions of the inertial and ambient sensors. The inertial_aligned and ambient_aligned directories show the tuples used on the DL framework presented.

The biggest regret of the article is that the model itself is limited to a fixed segment of 2 seconds, but the data set is about one minute per segment.

Most related works on video-based activity recognition provide architectures composed of DL models that operate on short segments, less than on-second-long, which may be further combined in different ways. We have adopted the approach of [29], which consists of simply averaging the predictions over several segments of the video during the test procedure. This policy has been adopted by other authors that address this modality [18,36]. For the other modalities (i.e., inertial and ambient sensors), we could do experiments with DL architectures that process the full-length data, but this would not be realistic for the application scenarios that we envisage.

We designed the proposed method for classification focusing on its possible applications on real-time AAL scenarios, especially considering the introduction of social robots. In this context, most expected behaviours of the artificial agent (e.g., actuators from the smart home or social robots) might be completed while the participant is still performing the activities, it is, before the activity is finished. This may be easily be envisaged as a requirement in several contexts. For example, a robot may need to bring the user’s glasses while he is reading the news - it would make little sense to do so after the user has stopped this activity. Besides, the dataset is composed of previously segmented activities, which is not realistic for real-world scenarios. It is important that the methods employed for classification can provide good predictions based on partial data. To highlight the reasons that led to the design of a framework focused on segment-wise classification, we have introduced Subsection 4.2.1.

The data set description of HWU/USP-MHAD lacks statistical information like Table 4, including average video length, minimum and maximum length.

Statistical information was provided in the new Figure 1.

After I downloaded the dataset found that: Neither the number of action types nor the number of videos in this dataset exceeds that of other datasets. Is it only part of it or is it all already?

The dataset made available is complete. We have a set of 16 participants performing 9 activities, which results on 144 instances. It may be worth highlighting that our recordings are significantly longer than those of the datasets more related to ours. This means that the total length exceeds those of most other multimodal datasets. A larger number of actions may be provided if fine-grained annotations are provided, for example by splitting an activity such as preparing a cup of tea in a series of shorter-length, less abstract actions such as taking a tea bag, turning on the kettle, and so on. This can be done according to the needs of any related research, by manually segmenting the video recordings and annotating the corresponding timestamps of the other modalities (see new content with respect to this issue in lines 494-502).

If you proposed a new data set, but immediately achieved very high accuracy (98.61%), that means this data set is not difficult enough. Consider adding some more complex actions, or combinations of actions as the challenging split of the data set.

Some activities have a sequence of more fine-grained actions (e.g., to prepare a sandwich, the participant had to place a plate on the board, take the ingredients from the fridge, take the bread from the cupboard, assemble the sandwich, and so on). Although such fine-grained annotations were not provided in the dataset, they may be introduced based on the videos, without the need of more data collection. This would comprise a different challenge that could be addressed in future work (see lines 494-502).

Nonetheless, there is still room for improvement regarding the modalities other than the videos, which is important for actual AAL applications, since video data might not always be available due to privacy issues. If those videos will be gathered by a robot’s camera, the positioning of the robot, which may not always be facing the inhabitants of the environment, may be another limitation. The very high accuracy of the video modalities may also serve to provide automatic annotations on semi-supervised settings for gathering data. An observation on this issue was introduced to the paper, lines 1013-1020.

  • Is it possible to replace all “depth videos” with a unified expression, such as RGB-D videos?

Our concern with this respect is that the RGB-D expression may refer to both RGB and depth data simultaneously, as provided by the RGB-D cameras, which was usually not the case when we talked about depth videos on our paper. Instead, we referred to the sequences of depth maps alone, without colour or intensity maps accompanying it.

We agree that using these expressions interchangeably could make the text less clear, hence we replaced those expression in the text so that “RGBD” is now used to refer only to the devices used to acquire this type of data, while “depth videos” is used to distinguish the sequences of depth maps from the “RGB videos”.

  • What is the main difference or contribution of the proposed dataset? Although in the paper, the difference is mentioned, what it can be driven or what it can be solved by this dataset. Please show the main contribution of this dataset.

A statement on this issue was included in lines 395-401, as quoted:

“The multimodal datasets presented in Section 2 provide data from different kinds of videos and inertial sensors, but they did not include data from ambient sensors. The main contribution of our dataset is introducing the data from the smart home devices synchronously with videos and inertial sensors. Also, we provided videos of either activities made of repetitive patterns, such as reading a newspaper, and more complex activities with long-term dependencies, such as preparing a sandwich. This makes our dataset more realistic regarding the set activities, with respect to what could be expected on an actual AAL scenario, when compared to the others.”

  • A new Deep Learning (DL) framework is designed, however, there is nothing about the structure of the new Deep Learning model.

The DL architectures are presented on Figure 6, and discussed throughout Subsection 4.2.3. Nevertheless, Table 5 was included for a clearer visualisation of the details of each model, and the surrounding text was adapted accordingly.

  • There is too much unnecessary description of other unrelated content that leads the article loss principal line. Please remove some unrelated content such as amount of description of other dataset.

The presentation of other datasets was shortened by reducing the amount of details and discussions on the text. Section 2 now spans through one page less than in the previous version of the paper.

  • Please show more contribution of the proposed dataset, and provide a meaningful application, not just segmentation, classification. The proposed dataset needs to help, support, drive some application development.

We introduced a new subsection regarding this issue (see Subsection 4.2.1), and also included considerations in the conclusion (see lines 1003-1012).

  • Line 491. Equation 1. The meaning of this symbol is not clear, consider redesigning this equation and explanation.

The explanation was rewritten in order to make it clearer (see lines 566-580), with redefinition of several symbols.

  • Figure 5. The description of ambient is a bit rough, the value at each moment should be different.

Changed sample (see Figure 7).

  • Figure 6. Lack of explanation for figure (c).

Explanation was introduced to the caption of Figure 8.

  • Wrong number in Table 4, line ChAirGest, column size.

Fixed (see Table 4).

  • Figure 3(d) It looks messy, consider removing these black borders or combining them into one chart.

The black borders were removed (see Figure 3).

I hope that future tasks can consider using longer time series for classification and using an unfixed camera.

Experiments on segments of data from different lengths and the proposal of a new data collection procedure were introduced as directions for future research (see lines 1013-1020).

Reviewer 3 Report

This paper presents a dataset, that will be made publicly available, of heterogeneous ADLs taken in home environments from multiple sensors, i.e., ambient sensors, wearable sensors, and robot mounted RGB and depth cameras. The authors evaluated their dataset by implementing two deep learning architectures, one used to process video data (i.e., optical and scene flows) and another one to process IMU data. Classification performances are compared with those obtained by using the UTD multimodal human action dataset (UTD-MHAD), showing results comparable, but not superior, to the state of the art.

The authors notably improved the manuscript with respect to the previously submitted version. Nonetheless, still, some shortcomings remain as listed below, which require to be addressed to make this paper worth publishing.

1) Since the presented dataset includes both depth and RGB data, why do the authors not consider skeleton information like in the related study [102] (Table 6)?

2) The related study [74] surpasses the authors’ findings without the use of skeleton information, as stated by the authors themselves. However, the explanation provided by the authors, in lines from 699 to 704, to justify this situation is neither clear nor sufficient. The authors should: 1) better argument relationships between segment-wise classification and scenarios with partial data, and, more important, 2) they should set up an experimental AAL scenario with partial data in order to demonstrate the superiority of their approach over the study [74].

SPECIFIC COMMENTS

3) In Table 4, the sensor modalities addressed by “BodyMedia” and “eWatch” should be clearly stated, otherwise the reader has to guess what they are (accelerometers, compass, both?)

4) Sections “Results” and “Discussion” are ambiguously separated. The results section should report only achieved results with tables, figures, etc., giving only basic information useful for their interpretation. Whereas all other discussions should be moved to the “Discussion” section.

5) Lines 706-708: The statement beginning with “In relation to the spatial and depth-based models, their results differ…” seems to refer to a subject belonging to the previous section! This is a very strange spread of content between sections. As highlighted above (4), this sentence should be preceded by the others concerning the discussion of the results and that currently appear in the "results" section.

Author Response

1) Since the presented dataset includes both depth and RGB data, why do the authors not consider skeleton information like in the related study [102] (Table 6)?

The relative positions of skeleton joints, whose computation is usually provided within the frameworks of RGB-D cameras, consist of structured information, which can be preprocessed to extract features useful for traditional, shallow classification methods. These handcrafted features may be particularly useful on more homogeneous datasets, such as the UTD-MHAD, although they would not necessarily provide results as good for datasets with more variability regarding the users’ movements, such as ours.

When designing our dataset, we were interested in providing a framework based on DL techniques. These architectures have shown to provide good results for video classification on highly unstructured scenarios, such as user content uploaded to Youtube. Such situations are closer to real-world scenarios that we aim to address in our research. We did not consider data from skeleton joints for our dataset because, to do this properly, we would have to rely on feature extraction strategies that would be out of the scope of the DL framework presented, which we believe does not make sense.

The goal of our paper was to provide a multimodal framework based not only on videos and inertial sensors, but also on ambient sensors, and doing this in a more realistic scenario (i.e., complex activities being performed on a monitored environment), which could only be achieved with the dataset that we collected. The experiments with the UTD-MHAD were included to situate part of our framework with respect to other architectures in the literature, rather than improving the overall accuracy for it. A paragraph on this issue was included in the paper (see lines 649-655).

2) The related study [74] surpasses the authors’ findings without the use of skeleton information, as stated by the authors themselves. However, the explanation provided by the authors, in lines from 699 to 704, to justify this situation is neither clear nor sufficient. The authors should: 1) better argument relationships between segment-wise classification and scenarios with partial data, and, more important, 2) they should set up an experimental AAL scenario with partial data in order to demonstrate the superiority of their approach over the study [74].

We designed the proposed methods for classification focusing on their possible applications on real-time AAL scenarios, especially considering the introduction of social robots. In this context, most expected behaviours of the artificial agent (e.g., actuators from the smart home or social robots) might be completed while the participant is still performing the activities, it is, before the activity is finished. This may be easily be envisaged as a requirement in several contexts. For example, a robot may need to bring the user’s glasses while he is reading the news - it would make little sense to do so after the user has stopped this activity. Besides, the dataset is composed of previously segmented activities, which is not realistic for real-world scenarios. It is important to note that the methods employed for classification can provide good predictions based on partial data. To highlight the reasons that led to the design of a framework focused on segment-wise classification, we have introduced a new subsection (see Subsection 4.2.1).

3) In Table 4, the sensor modalities addressed by “BodyMedia” and “eWatch” should be clearly stated, otherwise the reader has to guess what they are (accelerometers, compass, both?)

The eWatch is a device with a set of simple sensors: 3D accelerometer, audio, light and temperature sensors. However, by downloading the data, we observed that only the accelerometer data was present. We introduced this information to the table. The dataset webpage lacked information about the BodyMedia device. Considering that these recordings were present only for a few instances, we have removed this reference from the table.

4) Sections “Results” and “Discussion” are ambiguously separated. The results section should report only achieved results with tables, figures, etc., giving only basic information useful for their interpretation. Whereas all other discussions should be moved to the “Discussion” section.

All content related to discussions or comparisons presented at the “Results” section was merged with the text at the “Discussion” section, at the adequate positions and performing the adaptations needed.

5) Lines 706-708: The statement beginning with “In relation to the spatial and depth-based models, their results differ…” seems to refer to a subject belonging to the previous section! This is a very strange spread of content between sections. As highlighted above (4), this sentence should be preceded by the others concerning the discussion of the results and that currently appear in the "results" section.

This content was merged with the first discussions regarding the RGB and depth modalities, which now precedes it in the first paragraph of the “Discussion” section. Text adaptations were applied at the beginning of the sentence.

Reviewer 4 Report

The paper discusses two main contributions which are (1) the introduction of an activity recognition dataset and (2) the proposal of a deep learning "framework" that recognizes activities from videos, inertial sensors, and ambient sensors.

Concerning the first contribution: the proposed dataset might cover some existing gaps but I doubt that it is very needed by researchers who could rely on other existing datasets. I am also not sure whether the introduction of a new dataset could be considered a true contribution that could be compared to others in the field. In addition, it would be a good idea to come up with a shorter name for the dataset.

Concerning the second contribution: the used techniques are not new and the results showed that the developed approach could not beat the state-of-the-art. Also, the fact that the introduction of data from ambient sensors improved the results is an obvious and expected results that occurs whenever more data, especially from different sensor modalities, is introduced. I do not think anyone could argue against this since it is an established fact, therefore I cannot consider it a significant contribution.

Author Response

Concerning the first contribution: the proposed dataset might cover some existing gaps but I doubt that it is very needed by researchers who could rely on other existing datasets. I am also not sure whether the introduction of a new dataset could be considered a true contribution that could be compared to others in the field.

The multimodal datasets presented in Section 2 provide data from different kinds of videos and inertial sensors, but they did not include data from ambient sensors. The main contribution of our dataset is introducing the data from the smart home devices synchronously with videos and inertial sensors. Also, we provided videos of either activities made of repetitive patterns, such as reading a newspaper, and more complex activities with long-term dependencies, such as preparing a sandwich. This makes our dataset more realistic regarding the set activities, with respect to what could be expected on an actual AAL scenario, when compared to the others  (see lines 395-401).

In addition, it would be a good idea to come up with a shorter name for the dataset.

We have renamed the dataset from “HWU/USP-MHAD” to “HWU-USP activities dataset”, referencing it as simply “HWU-USP dataset” for most of the paper.

Concerning the second contribution: the used techniques are not new and the results showed that the developed approach could not beat the state-of-the-art.

The goal of our paper was to provide a multimodal framework based not only on videos and inertial sensors, but also on ambient sensors, and doing this in a more realistic scenario (i.e., complex activities being performed on a monitored environment), which could only be achieved with the dataset that we collected. The experiments with the UTD-MHAD were included to situate part of our framework with respect to other architectures in the literature, rather than improving the overall accuracy for it.

Even though, among the state-of-the-art methods that we have presented for the UTD-MHAD, only one of them showed overall accuracy higher than ours with only videos and inertial sensors [76]. It is important to note that the video-based method presented by them actually performed less accurate than ours (76.0%, against 84.8% of our approach), which means that the overall accuracy on their work benefited especially from the inertial-only model, which hit 90.3%, against 82.2% of ours. We included two rows to Table 7 to allow a single-modality discussion of these state-of-the-art results (see lines 949-953).

It holds to highlight that the inertial-based approach by [76] relies on a matrix representation which can only be elaborated with the full-length data, which is not possible in the context of our segment-wise approach. Our predictions for all modalities were based on two-seconds-long segments. We designed the proposed methods for classification focusing on its possible applications on real-time AAL scenarios, especially considering the introduction of social robots. In this context, most expected behaviours of the artificial agent (e.g., actuators from the smart home or social robots) might be completed while the participant is still performing the activities, it is, before the activity is finished. This may be easily be envisaged as a requirement in several contexts. For example, a robot may need to bring the user’s glasses while he is reading the news - it would make little sense to do so after the user has stopped this activity. Besides, the dataset is composed of previously segmented activities, which is not realistic for real-world scenarios. It is important that the methods employed for classification can provide good predictions based on partial data. To highlight the reasons that led to the design of a framework focused on segment-wise classification, we have introduced Subsection 4.2.1.

Also, the fact that the introduction of data from ambient sensors improved the results is an obvious and expected results that occurs whenever more data, especially from different sensor modalities, is introduced. I do not think anyone could argue against this since it is an established fact, therefore I cannot consider it a significant contribution.

It is important to consider that the different modalities of data considered compose a heterogeneous setting, with each sensor providing different representations to another. In this context, sensor fusion is not as trivial [27], and different techniques have been proposed to deal with this in the context of activity recognition [36,37,76]. In particular, the design of a fusion architecture has been pointed as a legitimate direction for research [28]. A statement on this issue was introduced along with the presentation of the second contribution in the Introduction (see lines 93-103).

In our paper, we proposed a DL-based fusion architecture for inertial and ambient sensor data. Before performing the experiments, there was no guarantee that this particular approach would lead to a result better than a similar neural network which relied only on inertial data, given that the ambient sensor data, less accurate, could introduce noisy features and lead to less accuracy. However, the proposed approach has shown to be successful.

The other fusion architectures presented were based on late combinations of output vectors, which might show a more predictable behaviour. Still, experiments are required in order to quantify the improvements that may be achieved by each combination of sensors. We not only provided such quantification by presenting the accuracies achieved by different combinations of sensors, but we also reported the confidences of the classifications throughout the segments of classification, considering different conditions (see Figures 11 and 12)

Round 2

Reviewer 1 Report

The authors taken into account my previous comments. 

Reviewer 3 Report

This paper presents a publicly available dataset of heterogeneous ADLs taken in home environments from multiple sensors, i.e., ambient sensors, wearable sensors, RGB and depth cameras. The authors evaluated their dataset using deep learning architectures. Classification performances are compared with those obtained by using the UTD multimodal human action dataset (UTD-MHAD).

The authors have made all necessary amendments to the previous reviewer’s comments; thus, the manuscript is now ready for acceptance.

Reviewer 4 Report

The authors addressed my major concerns and I am fine with the article being accepted. Good job!

This manuscript is a resubmission of an earlier submission. The following is a list of the peer review reports and author responses from that submission.

Round 1

Reviewer 1 Report

The idea to compare different datasets is interesting, but the paper should be organized.

The background provided in the introduction should be extended. The concept of human activity recognition should be explained in the introduction.

The section 2 should be focused on the datasets, and section 3 should be focused on the methods previously implemented.

Your section 5 should be merged with the description of the methods.

The results should be detailed, and the discussion should be another section.

The conclusions should be improved according to the discussion.

The following studies must be added:

[1] “RGB-D images for object segmentation, localization and recognition in indoor scenes using feature descriptor and Hough voting”, IEEE conference on applied sciences and technology, 2020. [2] “Depth Silhouettes Context: A new robust feature for human tracking and activity recognition based on embedded HMMs,” in Proceedings 12th IEEE International Conference on Ubiquitous Robots and Ambient Intelligence, pp. 294-299, 2015. [3] “Robust human activity recognition from depth video using spatiotemporal multi-fused features, Pattern recognition, vol. 61, pp. 295-308, 2017. [4] “Wearable Sensors for Activity Analysis using SMO-based Random Forest over Smart home and Sports Datasets”, IEEE ICACS conference, 2020. [5] “Depth Images-based Human Detection, Tracking and Activity Recognition Using Spatiotemporal Features and Modified HMM, Journal of Electrical Engineering and Technology, pp. 1921-1926, 2016. [6] “An Accurate Facial expression detector using multi-landmarks selection and local transform features,” IEEE ICACS conference, 2020. [7] “Dense RGB-D Map-Based Human Tracking and Activity Recognition using Skin Joints Features and Self-Organizing Map,” KSII Transactions on internet and information systems, vol. 9(5), pp. 1856-1869, 2015.

[1] “Vision-based Human Activity recognition system using depth silhouettes: A Smart home system for monitoring the residents, Journal of Electrical Engineering and Technology, 2019. [2] Development of a life logging system via depth imaging-based human activity recognition for smart homes,” in Proceedings of the International Symposium on Sustainable Healthy Buildings, pp. 91-95, 2012. [3] “Human actions tracking and recognition based on body parts detection via Artificial neural network,” IEEE International Conference on Advancements in computational sciences, 2020.. [4] Human activity recognition via recognized body parts of human depth silhouettes for residents monitoring services at smart homes, Indoor and Built Environment, Vol. 22, pp. 271-279, 2013. [5] “Wearable Sensors based Human Behavioral Pattern Recognition using Statistical Features and Reweighted Genetic Algorithm, Multimedia Tools and Applications, 2019. [6] Shape and motion features approach for activity tracking and recognition from Kinect video camera, in Proceedings 29th International Conference on Advanced Information Networking and Applications Workshops, pp. 445-450, 2015. [7] A depth video sensor-based life-logging human activity recognition system for elderly care in smart indoor environments, Sensors, vol. 14(7), pp. 11735-11759, 2014.

[1] “New shape descriptor in the context of edge continuity,” CAAI Transactions on Intelligence Technology, Vol. 4, no. 2, pp. 101-109, 2019. [2] “Three-stage network for age estimation,” CAAI Transactions on Intelligence Technology, Vol. 4, no. 2, pp. 122-126, 2019. [3] “Influence of kernel clustering on an RBFN,” CAAI Transactions on Intelligence Technology, Vol. 4, no. 4, pp. 255-260, 2019. [4] “Engine speed reduction for hydraulic machinery using predictive algorithms,” Int. J. Hydromechatronics, Vol. 2, no. 1, pp. 16-31, 2019. [5] “Analytical analysis of single-stage pressure relief valves,” Int. J. Hydromechatronics, Vol. 2, no. 1, pp. 32-53, 2019. [6] “A review on the artificial neural network approach to analysis and prediction of seismic damage in infrastructure,” Int. J. Hydromechatronics, Vol. 2, no. 4, pp. 178-196, 2019. [7] “WHITE STAG Model: Wise Human Interaction Tracking and Estimation (WHITE) using Spatio-temporal and Angular-geometric (STAG) Descriptors, Multimedia Tools and Applications, 2020.

and others... 

Reviewer 2 Report

  1. This manuscript’s motivation is great, that create multimodality data for human activity recognition in the smart home research field. It has two folds contributions: create HWU/USP-MHAD dataset and proposes a deep learning framework to perform experiments with this dataset. However, I don’t think these two contributions are enough.
  2. First, the HWU/USP-MHAD dataset is small, only 16 subjects with 9 activities, each person has about 20min data (total about 180min data). It is too small to use for deep learning model, especially for the activity classification task. Moreover, each activity data keeps the same starting points. While is good for training DL model and preventing overfitting, but this constraint setting lets this dataset not generalize and useful for real daily scene.
  3. Then, the proposed DL architecture has less innovation, that just simply uses CNN for extracting feature and LSTM for modeling sequence. And its comparison experiments performance on UTD-MHAD dataset (91.17% acc) is not well (compare to Wei el al. 95.6%).
  4. A typo, line 60, “wil” should be “will”.

Reviewer 3 Report

This paper presents a dataset, that will be made publicly available soon, as stated by the authors, of heterogeneous ADLs taken in home environments from multiple sensors, i.e., ambient sensors, wearable sensors, and robot-mounted RGB and depth cameras. The authors evaluated their dataset by implementing two deep learning architectures, one used to process video data (i.e., optical and scene flows) and another one to process IMU data. Classification performances are compared with those obtained by using the UTD multimodal human action dataset (UTD-MHAD), showing state-of-the-art results.

The paper is very interesting and technically sound, and surely it will be of interest to the readers of this journal; however, there are some aspects that are omitted or underemphasised in the report that could be improved as listed below.

- The process mentioned by the authors of converting a depth frame into 3-channel RGB frames (line 403-404, p. 12) should be described more accurately and explicitly, since it is not so straightforward. This process, additionally, should also be considered in the discussion of Table 5 results, comparing the differences in performance between optical flow VS scene flow. Indeed, in this reviewer's opinion, it is not enough to say that "the differences in performance between the optical flow and the scene flow models might have been due to differences inherent to the algorithms employed" (lines 519-520, p.16). These differences should be assessed and carefully considered also in consideration of the aforementioned conversion process from depth to RGB channels.

- It would be interesting to see performance results, in addition to those reported in Table 5, when outputs from the optical flow are fused to those from scene flow (Optical flow + Scene flow) and when all outputs are fused together (Optical flow + Scene flow + IMU).

- There are many formatting mistakes around the manuscript that should be addressed, mainly regarding acronym definition and referencing style. All acronyms should be defined the first time they are used (e.g., IMU is not defined, RGBD, and others). Many citations around the manuscript are reported without the brackets (e.g., line 92 p. 3, lines 100-101 p. 3, line 116 p. 3, etc., many others are present).

- In Table 4, please provide the measurement unit of the column "Size"; is it in number of frames, time duration (minutes, seconds), or what?

- In section 1 (Introduction), all methodological statements, such as those at p. 2 beginning with "All activities were performed by 16 participants…" and " All of the participants were healthy…" and "Our modalities include switch sensors…" and "In addition two wearable IMUs were placed…" and so on for other similar statements, should be moved to section 4 (The HWU/USP-MHAD Dataset).

- Manufacturers of used devices are not cited, such as "Microsoft Kinect" (line 186), "TIAGo robot" (line 311), "Orbbec Astra" (line 314), "MetaMotionR" (line 320), but others might be present around the manuscript.

Reviewer 4 Report

The problems connected with the ageing population all over the world have become more and more severe. Different sensor systems have been proposed for monitoring the functional abilities in elderly and for detecting their functional decline. Ambient Assisted Living (AAL) systems have become a very popular strategy to adapt the patients’ living environments to their specific needs, which improves life quality and optimize healthcare resources.

The monitoring and analysis activities of daily living (ADL), through the application of various technological solutions, emerges as a reliable approach for adequate assessment of the condition of adults. An indispensable step in the development of such complex innovative technological solutions is their setup and testing using databases, including a wide range of signals, parameters, images and other indicators characterizing ADL.

The aim of the authors is to synthesize and study a complex dataset, focusing on 9 activities performed in a kitchen environment during breakfast. All data are captured in a heterogeneous, sensory environment comprised of a: smart home system, wearable sensors and a robot equipped with RGB and depth cameras.

A deep learning framework based on convolutional neural networks modules is provided, allowing to receive reference results in classification of the labelled activities from the dataset. In accordance with the established practice, a similar approach is used in the analysis of another widely used database - UTD-MHAD. The obtained results show that the developed by the authors approach, trained and tested with the newly synthesized database, are comparable with the best published results by other authors in similar tests with UTD-MHAD.

The article is well structured and demonstrates deep knowledge of the problematics and logical sequence in explanations. The reference sources correspond to the content and are cited on the appropriate places in the text. Results and directions for future work are properly presented in the final part.

Some comments questions and suggestions:

  • The use of atypical abbreviations in the title and the abstract, without their depiction, is considered as unacceptable and does not contribute to the correct understanding of the content of the article before careful reading.

A more detailed comments are needed regarding to the influence of the differences in the databases on the accuracy of the feature extraction and temporal modelling, in particular: (1) – in UTD-MHAD only one inertial sensor worn on the subject's right wrist or on the right thigh, while in HWU/USP-MHAD two sensors are used - one sensor on the subject’s dominant arm and second sensor placed on the subject waist; (2) - The processed images in UTD-MHAD are only in full face, while in HWU/USP-MHAD the processed images are in profile and in full face.

I want to draw the authors’ attention to the following typo inaccuracies:

Ln 45 – “…may may..”

Ln 135 – “Liciotti et alLiciotti et al.”

Ln 323 – “..”subjects wrist”) should be “subjects waist”.

Ln 440 – “and and”

Reviewer 5 Report

I do not think the contributions made by this work are enough to publish it. The main contribution is the dataset, which should be published on its own, not through a research paper. As for the classification algorithm, which could serve as an actual publishable contribution, it failed to overperform the state-of-the-art, so it is hard for me to be convinced that this contribution would help others working on classifying activities.